

# BREAD: Branched Rollouts from Expert Anchors Bridge SFT & RL for Reasoning

**Xuechen Zhang***
University of Michigan
zxuechen@umich.edu

**Zijian Huang***
University of Michigan
zijianh@umich.edu

**Yingcong Li**
University of Michigan
yingcong@umich.edu

**Chenshun Ni**
University of Michigan
nichensh@umich.edu

**Jiasi Chen**
University of Michigan
jiasi@umich.edu

**Samet Oymak**
University of Michigan
oymak@umich.edu

## Abstract

Small language models (SLMs) struggle to learn complex reasoning behaviors, especially when high-quality traces are scarce or difficult to learn from. The standard training approach combines a supervised fine-tuning (SFT) stage, often to distill capabilities of a larger model, followed by a reinforcement learning (RL) stage such as Group Relative Policy Optimization (GRPO). In this paper, we investigate the fundamental limitations of this SFT + RL paradigm and propose methods to overcome them. Under a suitable theoretical model, we demonstrate that the SFT + RL strategy can fail completely when (1) the expert's traces are too difficult for the small model to express, or (2) the small model's initialization has exponentially small likelihood of success. To address these, we introduce BREAD: a GRPO variant that unifies the SFT and RL stages via partial expert guidance and branched rollouts. When self-generated traces fail, BREAD adaptively inserts short expert prefixes/hints, allowing the small model to complete the rest of the reasoning path, and ensuring that each update includes at least one successful trace. This mechanism both densifies the reward signal and induces a natural learning curriculum. BREAD requires fewer than 40% of ground-truth traces, consistently outperforming standard GRPO while speeding up the training by about $3\times$. Importantly, we demonstrate that BREAD helps the model solve problems that are otherwise unsolvable by the SFT + RL strategy, highlighting how branched rollouts and expert guidance can substantially boost SLM reasoning.

## 1 Introduction

Over the past few years, we have witnessed a significant push toward enhancing language model reasoning, which has led to highly capable frontier models such as OpenAI o1 [17], Gemini 2.5 [19], and DeepSeek R1 [13]. These models can generate longer chain-of-thought (CoT) traces and utilize more test-time compute to tackle challenging tasks [24]. Despite these innovations, reasoning with small language models (SLM) remains a challenge. For instance, the large DeepSeek-R1 [13] has 671B parameters in total whereas the SFT-distilled model sizes range from 1.5B to 70B, and their performance substantially degrades at the 1.5B model (see Table 5 in [13]).

This work studies optimization strategies to enhance SLMs, with emphasis on reasoning tasks. Two popular optimization strategies for training LLMs are supervised fine-tuning (SFT) and reinforcement learning (such as GRPO or proximal policy optimization). Often, an SFT phase is employed, followed by an RL phase. While this two-stage procedure has found success, for SLMs, long-context reasoning

---

*These authors contributed equally to this work. Correspondance to {zxuechen,zijianh,oymak}@umich.edu

39th Conference on Neural Information Processing Systems (NeurIPS 2025).

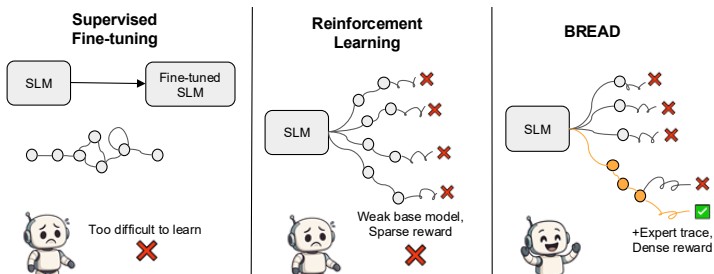

Figure 1: High-level overview of approaches. In existing training methods like supervised fine-tuning (left), high-quality reasoning traces produced by LLMs are often too complex for SLMs to imitate, so they deliver little benefit and can even hurt SLM reasoning capability. Since the subsequent RL phase starts from this weak starting point, the two-stage SFT+RL procedure often fails. In reinforcement learning (center), when the initial policy generates incorrect traces, the rewards are sparse, causing slow or ineffective learning. We propose BREAD (right), which uses part of an expert trace as an anchor, then generates additional rollouts that branch from its intermediate episodes. These branched trajectories provide denser, higher-quality feedback, helping SLMs learn robust reasoning strategies. Each dot represents a single episode, and the yellow trajectory is the expert trace.

problems can pose unique challenges due to the misalignment between the expert and student models and potentially sparse rewards.

For example, consider a scenario in which each token generated by an expert/teacher model requires the smaller student model to produce $K$ intermediate tokens to express it. In other words, the expert thinks and outputs $K$ steps ahead, from the student's point of view. In practice, this situation can arise when the expert model is a $\times K$ deeper/looped version of the student. Naturally, such expert traces might be too challenging for the small model to learn from[2]. On the other hand, success of the RL phase often relies on good supervised initialization during the SFT phase. In Section 2.1, we provide a mathematical setting capturing this intuition and demonstrate that SFT + RL can fail for small models (see Figure 3), especially on difficult problems.

To address the difficulties of small models in learning from complex traces during fine-tuning, we propose our algorithm, Branch Rollouts and Expert Anchors for Densified rewards, depicted in Figures 1 and 2. BREAD gracefully integrates the SFT and RL phases by anchoring the optimization process with the expert traces, while allowing the small base model to acquire progressively more flexibility as it becomes a stronger problem solver. Assuming that an expert trace is available (e.g. by querying a large expert model), BREAD updates the model with a correct trace; however, its traces are progressively more self-generated.

**Contributions:** Our specific contributions are as follows:
- **Methodology:** We introduce BREAD: a GRPO variant that generalizes conventional SFT and RL phases through the use of branched rollouts to automatically adapt to the problem difficulty. BREAD induces a learning curriculum along the reasoning trace, thereby densifying the reward signal and tackling compositional tasks step-by-step. We investigate the theoretical properties of BREAD in a suitable student-expert setting. In this setting, we establish how the expert model's trace can be uninformative to the student and how the subsequent RL phase can fail due to sparse rewards. In contrast, we show that BREAD can solve the target problem efficiently (low training cost) and concisely (short trace length).
- **Empirical impact:** Our experimental results show that BREAD matches or surpasses SFT + vanilla GRPO while using $\approx 20\%$ of the correct trace tokens. By reducing both the number of rollouts and the total optimization steps, BREAD reduces the overall training compute by $\approx 75\%$ relative to vanilla GRPO. Additionally, we construct a slice of difficult problems and demonstrate that BREAD, when trained on this set, achieves substantially higher accuracy compared to SFT+GRPO, corroborating its fundamental benefits in solving difficult problems that are otherwise unsolvable by SFT+GRPO. Finally, we provide an empirical study of how branched rollouts from expert hints (i.e., using only parts of the expert trace) densifies the reward signal.

The rest of the paper is organized as follows: Section 5 dicusses the related work on language model reasoning and traditional RL methods. Section 1 explains our algorithm BREAD, which also contains the observations inspiring our algorithm design. Section 4 presents and discusses our main experiment results to demonstrate the effectiveness of BREAD. Section 6 concludes the paper and discusses the limitation and future directions to improve in this domain.

---

[2]In practice, SFT+RL can work well for much of the dataset but might fail on a subset of a difficult problems. See Section 4.1 for empirical evidence and evaluations.

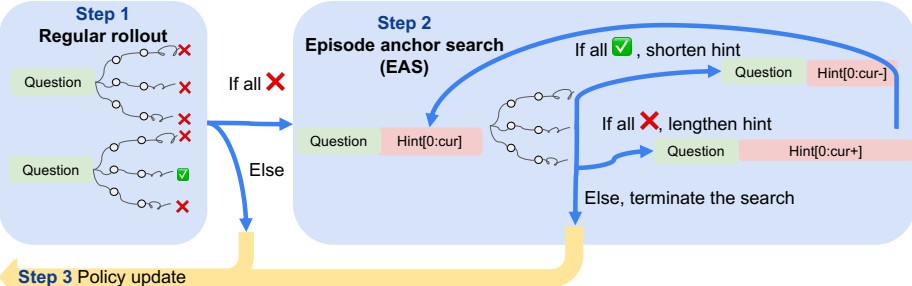

Figure 2: Workflow of BREAD. (1) **Regular rollout**: Given a question $Q$, sample a group of rollouts. If the sampled rollouts contain correct reasoning trace, use this group of rollouts to do the **policy updates**. Otherwise, go to step (2) **Episode anchor search**: Starting with the whole expert trace (provided by the ground truth or from the correct responses generated by LLMs) as the search space for potential suitable hints, construct a hint using the first half of the expert trace, append it to the question $q$, and sample a group of rollouts. If all the rollouts are correct, shorten the hint to contain fewer episodes and repeat the process; if all rollouts are wrong, lengthen the hint; otherwise, use the current rollouts to do the **policy update**.

## 2 Proposed Method: BREAD

In this work, we propose the **B**ranched **R**ollouts and **E**xpert **A**nchors for **D**ensified RL (BREAD) algorithm. We will first describe the algorithm at a high level, followed by a mathematical toy model example (Section 2.1), and the key observations (Section 3.1, Section 3.2) that support its design.

In BREAD, for each question $q$ paired with the answer $a$, the workflow of BREAD is shown in Figure 2 and proceeds as follows:

1. **Regular rollout:** Sample a group of $G$ rollouts $\{o_i\}_{i=1}^{G}$.
2. **Episode anchor search:** If the success rate of the initial group is low (e.g. lower than a threshold), do a binary search to find a short hint $\rho$ that contains the "Expert Anchor" in the expert solution. "Expert Anchor" means the success rate of a new sampled output group $\{o_i'\}_{i=1}^{G}$, resulting from the question appended with the hint from the expert trace $(q, \rho)$, is within a pre-defined range.
3. **Policy updates:** Optimize the policy via the following objective:

$$\mathcal{J}_{\text{BREAD}}(\theta) = \mathbb{E}_{(q,\rho,a)\sim\mathcal{D},\{o_i\}_{i=1}^{G}\sim\pi_{\text{old}}(\cdot|(q,\rho))}$$

$$\left[ \frac{1}{G}\sum_{i=1}^{G}\frac{1}{|o_i|}\sum_{t=1}^{|o_i|}\left( \min\left( r_{i,t}(\theta)\hat{A}_{i,t}, \text{clip}\left( r_{i,t}(\theta), 1-\varepsilon, 1+\varepsilon \right)\hat{A}_{i,t} \right) - \beta D_{\text{KL}}(\pi_\theta||\pi_{\text{ref}}) \right) \right], \quad (1)$$

where

$$r_{i,t}(\theta) = \frac{\pi_\theta(o_{i,t}|q,\rho,o_{i<t})}{\pi_{\text{old}}(o_{i,t}|q,\rho,o_{i<t})}, \quad \hat{A}_{i,t} = \frac{r_i - \text{mean}(\{R_i\}_{i=1}^{G})}{\text{std}(\{R_i\}_{i=1}^{G})} \quad (2)$$

The details of the algorithm can be found in Algorithm 1. Next, we will discuss the key observations that motivate the design of BREAD.

### 2.1 Theoretical Insights into SFT, Reinforcement Learning, and BREAD

**Challenges of SFT:** It is known that a $L$ times deeper LLM can internally simulate $L$ chain-of-thought steps of a smaller LLM [31]. This implies that the expert model can generate a dense reasoning trace which necessitates a $L\times$ longer simplified trace for the student model to digest via SFT. Otherwise, the student model may fail to learn from SFT as we will elaborate further below. Recent work [22] makes related empirical observation that SLMs can struggle to learn from strong reasoners.

**Insights from compositionality:** Reasoning problems are inherently compositional hence we need a chain-of-thought [39] to solve such problems. Suppose the problem contains $T$ steps/subtasks to solve, each with successful completion probability of $\epsilon$ for the base student model. Then, we receive a final trajectory-level correctness reward with probability of $\epsilon^T$ assuming steps are independently completed. In contrast, by branching out from the expert trace at step $T-\tau$, BREAD only needs to complete the $\tau$ downstream steps, increasing the success to $\epsilon^\tau$. Notably, by increasing $\tau$ from 1 to $T$, BREAD facilitates curriculum learning by tackling individual problem steps one at a time requiring only $\Omega(1/\epsilon)$ samples to attain success per step. Below we further formalize this discussion under a Markov chain model.

**Markov chain model:** Let us model the expert and student models through Markov chains as follows: Imagine a Markov chain over the States $\{0, 1, 2, \ldots, K\}$. We envision that the student model is

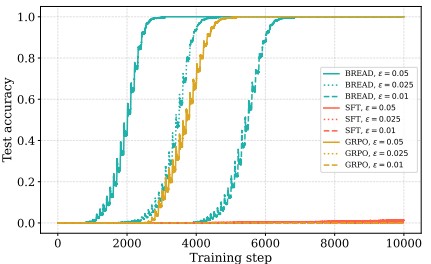 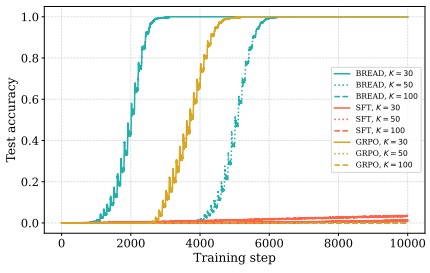

(a) Varying $\epsilon$ – Initialization difficulty  (b) Varying $K$ – Number of states

Figure 3: We compare SFT, GRPO, and BREAD according to the Navigation Task described in Section 2.1. $K$ is number of states in the Markov chain whereas $\epsilon$ is the probability of transition to *non-favorable* states. Our toy model reveals settings where BREAD can succeed while SFT or GRPO completely fail.

forbidden from implementing certain state transitions. Specifically, its connections are local and consecutive state transitions obey $|x_{t+1} - x_t| \leqslant d$ where the agent reaches state $x_t$ at time $t$, and $d$ denotes its maximum allowable jump distance of the small model. We consider the learnability of the following navigation task by the student.

> **Navigation Task:** Start a trace from State 0. Obtain a reward of 1 upon reaching State $K$.

Suppose the expert model solves the *Navigation Task* by generating the trace $\mathcal{T}_e = [x_0 = 0, x_1, x_2, \ldots, x_{T-1}, x_T = K]$ where $x_t \in \{1, \ldots, K-1\}$ for $1 \leqslant t \leqslant T-1$. During SFT, the student Markov chain can only learn from $\mathcal{T}_e$ when the expert transitions $(x_t, x_{t+1})$ are learnable, i.e., $|x_t - x_{t+1}| \leqslant d$. If no transition satisfies this, the small model cannot learn from SFT.

Without a good SFT-induced initialization, pure RL is known to suffer from sparse rewards [37]. In the context of the Navigation Task, once SFT fails, the student model can only improve through vanilla RL if it can generate a trace that successfully runs from $0$ to $K$ to earn a nonzero reward. In contrast, if the student employs BREAD, rather than starting from $0$, it can create a rollout starting from $x_{T-1}$. Thus, it only needs to complete a much easier task starting from $x_{T-1}$, earning a reward by reaching the final state $x_T = K$. In the context of compositionality, generation of traces from $x_{t-1}$ to $x_t$ corresponds to solving $t$'th subtask.

To proceed, we have experimented with the Navigation Task controlled by the parameters number of states $K + 1$, the student model's jump capacity $d$, and initialization quality $\epsilon$. Specifically, for each state $i$, the student model can jump to a neighbor $j \neq i, |i - j| \leqslant d$ with probability at most $\epsilon$. Hence, larger the $\epsilon$, the more model struggles under trace length constraint. Figure 3 demonstrates how optimization of SFT and GRPO can suffer as a function of $K$ and $\epsilon$ respectively. Importantly, we also display the performance of BREAD which exhibits a much more favorable performance. The reader is referred to the supplementary material for full experimental details. To proceed, we provide a concrete theoretical analysis of this discussion in the next section.

## 2.2 Formal Guarantees under a Random Walk Model

**Setting and assumptions:** The expert trace starts at State 0 and ends at State $K$ by following the trajectory $x_0 = 0, x_1, \ldots, x_T = K$. We assume that $c \cdot K/T \geqslant x_{t+1} - x_t \geqslant K/cT$ for some $c \geqslant 1$ and all $t$, and that the interval $[K/cT, cK/T]$ contains at least one positive integer. In words, the expert model has a jump size of $\Omega(K/T)$.

We assume that the student model is initialized as a *symmetric random walk*. That means, it can only move to left or right neighbor (jump size $d = 1$ and $\epsilon = 0.5$) and it is allowed to visit negative states. Thus, optimizing the student model is same as choosing optimal neighbor transition probabilities for a random walk. In our context, the ideal transition rule is always moving to the right (from $i$ to $i + 1$).

Finally, we enforce a maximum trace length of $L$ and, during RL, the model only receives reward if the trace successfully reaches state $K$ within $L$ steps. With these, we have the following lemma that establishes that SFT on the expert trace followed by reinforcement learning cannot improve the student model with high probability unless $L \geqslant \Omega(K^2)$.

**Lemma 1 (Failure of SFT+RL)** *Suppose the student model is a Markov chain with $d = 1$ initialized as a symmetric random walk. Suppose the expert trace $(x_t)_{t=1}^T$ always jumps two or more (i.e. $K >$*

$c \cdot T$). *Given the expert trace and at most $N \leqslant e^{K^2/4L}$ traces generated during the RL phase, SFT and GRPO training have no impact on the student model with probability at least $2e^{-K^2/4L}$.*

In words, this theorem states that the student is not able to learn from expert due to limited expressivity and it is not benefiting from the RL phase due to never obtaining nonzero reward. This failure happens unless the trace is long in the sense that $L \geqslant \Omega(K^2)$. Indeed, this is the time it takes for a symmetric random walk (initial student model) to hit State $K$.

Our next result proves how BREAD can overcome this bottleneck and address this bottleneck while conforming to a shorter trace length. To this aim, we examine a variation of BREAD that relies on a simple memorization of successful traces.

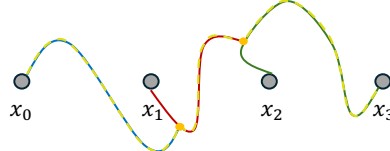

**Algorithm: BREAD variant.** The algorithm starts from the right most end of the expert trajectory and incrementally learns the correct completion. Concretely,

Figure 4: Depiction of the trajectory stitching argument for $T = 3$. Short subtraces (blue, red, green) are those generated at each round. The dashed yellow line are the final trace that maps $x_0$ to $x_T$.

• During the first round, create IID rollouts of length at most $L$ starting from the prefix $(x_t)_{t=0}^{T-1}$. Once a reward of 1 is achieved, record the generated suffix $S_1 = (x_t')_{t \geqslant T}$.

• At round $\tau \leqslant T$, create IID rollouts of length at most $L$ starting from the prefix $(x_t)_{t=0}^{T-\tau}$. Terminate a rollout once it hits a state $s_{\text{hit}}$ in $S_{\tau-1}$ and complete the trace by replaying $S_{\tau-1}$ from $s_{\text{hit}}$. Declare success if the total length is at most $L$ and set $S_\tau$ to be the union of generated rollout and $S_{\tau-1}$ continued from $s_{\text{hit}}$.

This algorithm incrementally learns sub-trajectories and stitch them to eventually obtain a full trajectory $S_T$ that solves the entire problem. This is depicted in Figure 4. During inference, the model simply replays the final memorized trace $S_T$ (yellow dashed curve in Figure 4).

We have the following theorem that establish the sample complexity of this BREAD variant.

**Theorem 1 (Success of BREAD)** *Suppose $L \geqslant 5c^2 K^2/T$ and assume BREAD generates $t$ rollouts at each round $1 \leqslant \tau \leqslant T$. Then, with probability at least $1 - Te^{-t}$, BREAD succeeds in all $T$ rounds in creating $S_\tau$ while never violating the maximum trace length $L$. Thus, BREAD memorizes a successful student trace $S_T$ with length at most $L$ with the same probability.*

A few remarks are in place: First, when $L \propto K^2/T$, BREAD succeeds by using a total of $\Omega(T \log(T))$ traces over $T$ rounds whereas SFT+RL from Lemma 1 fails with probability at least $1 - 2e^{-\Omega(T)}$ confirming our intuition on reward densification benefits. Secondly, suppose $T \propto K$ i.e. the expert model has constant jump size. In this case, BREAD solves the problem in linear time as it requires $L \gtrsim O(K)$. In contrast, SFT+RL requires $L \gtrsim O(K^2)$ to solve the problem. In the next section, we discuss how these theory and insights are in line with the SFT and RL performance on real reasoning tasks with state-of-the-art models.

## 3 Empirical Insights into BREAD

### 3.1 Limitations of SFT and Reinforcement Learning for SLMs

**RL-only evaluations:** We begin by assessing how well RL works with SLMs in isolation. Experiments with vanilla GRPO [33], displayed in Figure 5, show that it merely sharpens the capabilities the model already exhibits. GRPO relies on sampling a mix of good and bad traces. But when a small base model fails to produce any high-quality trace—***nearly half of the queries*** in our evaluations in Figure 5a—learning stalls due to lack of reward signals (Figure 5b). Related limitations are noted for RL with Verifiable Rewards of [42], which struggles to elicit fundamentally new reasoning patterns.

**SFT-only evaluations:** SFT can introduce new knowledge into the model. However, as discussed earlier, traces generated by much stronger models can be too complex for SLMs to learn resulting in poor initialization for RL. To demonstrate this, we start from Qwen2.5-1.5B-Instruct and Qwen2.5-3B-Instruct base models and fine-tune them with 1000 difficult questions, paired with reasoning traces generated by Gemini Thinking Experimental [11] and Deepseek-R1 [13], following [35]. These reasoning traces were previously shown to improve the reasoning capability of large models such as Qwen2.5-32B-Instruct and Qwen2.5-14B-Instruct by [24]. However, in our experiment results shown in Table 1, when SFT was performed on small models, ***accuracy actually decreases***. For example,

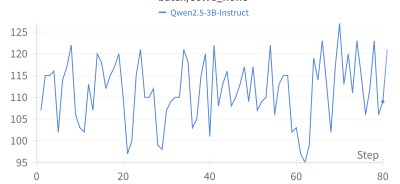

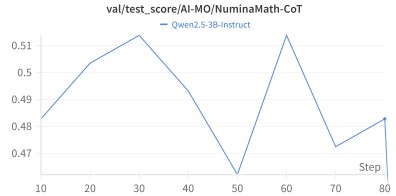

(a) Number of samples where all 8 rollouts fail  (b) Validation accuracy over training steps

Figure 5: Issues with vanilla GRPO on SLMs. Among 256 samples in a batch, for nearly half of them, the base model's 8 rollouts produce no correct trace. The absence of reward inhibits further performance gains and validation accuracy stalls.

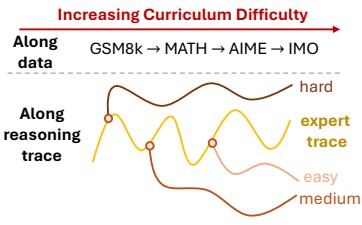

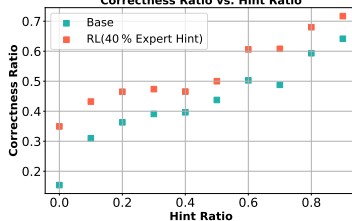

(a) Illustration of hint ratio (how much of the expert trace is provided) and curriculum learning.  (b) Accuracy for different ratio of hints, appended to the inference query.

Figure 6: (a) Math benchmarks may show difficulty tiers, but ranking individual problems for curriculum training is still challenging. Instead, BREAD can adaptively adjust the difficulty by automatically choosing the branching point. For easier questions, it would provide no or shorter hints (smaller hint ratio). (b) Model accuracy increases as longer hint is appended to the query. Dataset is the first 200 questions of NuminaMath-CoT. The blue dots are the base model (Qwen2.5-3B-Instruct), and the orange dots are RL finetuned using traces containing 40% hints.

accuracy dropped from 0.257 to 0.177 on the GPQA dataset when the 1.5B parameter model was fine-tuned on S1K traces, and dropped even further to 0.121 when fine-tuned on the more verbose S1K-1.1 data. This corroborates our central intuition: when traces used for SFT exceed an SLM's learning capacity, they can hurt the model performance rather than helping.

These findings—that for small models, RL alone, or SFT + RL, are insufficient—motivate BREAD which uses expert traces directly in the RL phase instead of relying on SFT to learn them first.

| Dataset | Qwen-1.5B-Instruct | | | Qwen-3B-Instruct | | |
|---|---|---|---|---|---|---|
| | Base | + SFT (S1K) | + SFT (S1K-1.1) | Base | + SFT (S1K) | + SFT (S1K-1.1) |
| MATH [15] | 0.494 | 0.426 | 0.408 | 0.624 | 0.616 | 0.630 |
| GPQA [28] | 0.258 | 0.177 | 0.121 | 0.369 | 0.247 | 0.288 |

Table 1: Accuracy of small models can decrease after SFT. The base SLMs are Qwen2.5-1.5B-Instruct and Qwen2.5-3B-Instruct and the datasets used for SFT are S1K and S1K-1.1 [24], with responses generated by Gemini Flash Thinking API [12] and DeepSeek-R1 [13] respectively.

## 3.2  Why BREAD: Expert Traces Provide Critical Guidance and Curriculum for RL

While Section 3.1 shows that an SLM can struggle to imitate traces generated by powerful LLMs, we posit that *the SLM can still understand them well enough to extract useful information*, namely by using part of the complex traces as hints to lower the problem difficulty. To illustrate this, we define the "hint ratio" as the fraction of the expert trace used (in terms of number of episodes), as illustrated in Figure 6a. A higher hint ratio means more guidance for the model. In Figure 6b, we plot the model accuracy for different hint ratios, where the hint is appended to the inference query. We can see that our intuition is correct: even providing part of the hint in a very simple way (by appending to the inference query) helps improve accuracy. In contrast, Figure 7 shows that SFT only on the same traces gets very limited improvement and can even hurt. BREAD is motivated by this observation and uses expert hints in a more sophisticated way, by integrating them directly into the RL create denser rewards.

We can also view BREAD through the lens of curriculum learning. As shown in Figure 6a, as we branch out from the expert trace earlier, solving the problem becomes more challenging. The Episode Anchor Search (EAS) step in BREAD automatically adjusts the branching point, producing a self-paced curriculum. In this sense, BREAD serves as a generalized curriculum-learning framework

which (1) allows us to utilize SFT data within RL, (2) adapts the optimization to the problem difficulty, and (3) generates dense rewards by creating a curriculum along the reasoning trace. By branching along the expert trace at adaptive cut-points, BREAD automatically surfaces the most informative training samples and tunes their difficulty on the fly. This relieves us from hand-crafting a data-level schedule and lets the model discover its own curriculum.

A final surprising observation from Figure 6b is that conditioning RL training on questions with partial hints not only improves performance on hinted queries (e.g., orange dots with hint ratio 0.3 and above), but also improves accuracy on questions without any hints (i.e., orange dot with hint ratio of 0). This highlights the model's ability to generalize from partial short traces to full long traces.

---

**Algorithm 1 BREAD**: GRPO with Branched Rollouts and Expert Anchors
(see Algorithm 2 in Appendix for full version.)

---

**Require:** Dataset $\mathcal{D}$, current policy $\pi_\theta$, sampling number $G$, a list of keywords for splitting episodes $[w_0, w_2, ..., w_J]$
1: **procedure** BREAD($\mathcal{D}, \pi_\theta, G, [w_0, w_2, ..., w_J]$)
2:     **for** step $= 1, 2, ..., N$ **do**
3:         Sample a batch $\mathcal{D}_b$ from $\mathcal{D}$
4:         Update the old policy model $\pi_{\text{old}} \leftarrow \pi_\theta$
5:         **for** each question and expert solution pair $(Q, S)$ in $\mathcal{D}_b$ **do**
6:             $[R_1, R_2, ..., R_G], \rho \leftarrow$ EAS($Q, S, \pi_\theta, G, [w_0, w_2, ..., w_J]$)     ▷ Episode Anchor Search (EAS)
7:             Add the $(Q, \rho), R_i$ to the buffer.
8:         **end for**
9:         For each $(Q, \rho), o_i$ in the buffer, compute $\hat{A}_{i,t}$ for the $t$-th token of $o_i$     ▷ Equation (2)
10:    **end for**
11:    **for** iteration $= 1, 2, ..., T$ **do**
12:        Update the policy model $\pi_\theta$ by maxmizing the BREAD objective     ▷ Equation (1)
13:    **end for**
14:    **return** $\pi_\theta$
15: **end procedure**
16: **procedure** EAS($Q, S, \pi, G, [w_0, w_2, ..., w_J]$)
17:    $[R_1, R_2, ..., R_G] \leftarrow \pi(Q)$     ▷ Sample $I$ responses for question $Q$ with current policy $\pi$
18:    $p_{\text{correct}} \leftarrow$ compute_correct_probability($[R_1, R_2, ..., R_G], S$)
19:                          ▷ compute correct probability based on responses and the gold solution
20:    **if** $0 < p_{\text{correct}}$ **then**
21:        **return** $[R_1, R_2, ..., R_G], NA$
22:    **else**
23:        $[e_0, e_1, ..., e_K] \leftarrow$ split_episodes($S, [w_0, w_2, ..., w_J]$)
24:        **return** BINARY_SEARCH_AND_GENERATE($Q, S, \pi, G, [e_1, e_2, ..., e_K], 0, K$)
25:    **end if**
26: **end procedure**

---

## 4 Experiments

**Baseline algorithms.** We evaluate the effectiveness of our method BREAD (as described in Algorithm 1) on mathematical tasks, which can be easily adapted to other reasoning tasks such as coding, commonsense reasoning, etc. We compare with the following baselines:

- GRPO: Standard GRPO.
- SFT: Standard supervised fine-tuning on the full training set. We denote SFT on various data splits as SFT(X), e.g. SFT(full) for the whole dataset, SFT(difficult) for the hardest subset, SFT(random) for a size-matched random subset, and SFT(selected) for the EAS-chosen subset. The precise discussion is detailed with the experiments results.
- SFT + GRPO: This means SFT is run followed by GRPO. In other words, GRPO continues from the final checkpoint of the same SFT run. We denote GRPO fine-tuning initialized from an SFT model trained on data split X as SFT(X) + GRPO.
- GRPO w/ Expert Trace: During GRPO training, the entire expert trace is injected as an additional rollout for all queries. This is a strong baseline as it contains the expert. Details in Appendix C.

**Training settings.** We adopt the verl framework [34] for training. We utilize the Adam optimizer [20] with a constant learning rate of $1 \times 10^{-6}$. For rollout, the prompt batch size is 256 and we sample 8 responses for each prompt. For training, the mini-batch size is 64. To find the appropriate branching point during the Episode Anchor Search (EAS) step of BREAD (Section 1), we first split the expert

trace into sentences (on easier datasets like MATH [15], where the expert traces are generally short) or into paragraphs (on harder datasets like NuminaMth-CoT [21] and OpenR1-Math-220K [9]). Then, we aggregate the split partitions into $K = 10$ episodes evenly, and proceed with binary search. See details in Appendix B.3.

### 4.1 Main Results

**BREAD outperforms all baselines in terms of test accuracy.** Figure 7 plots the training curves of all methods on the NuminaMath-CoT benchmark, starting from the Qwen-2.5-3B-Instruct base model. In Figure 7a, we visualize the test accuracy against training steps for BREAD and the baseline methods. BREAD outperforms all the baselines. Let us discuss each baseline in turn. BREAD improves final accuracy by more than 15% over vanilla `GRPO`. SFT can offer a stronger starting point for RL, which we can see by comparing the `SFT` and `SFT+GRPO` curves, but BREAD is still better. (Note that training SFT for more iterations does not further improve performance, see Appendix B.1.) `SFT(Difficult)` represents SFT trained on the hardest questions from NuminaMath-CoT, obtained by sorting samples by solution length. Intuitively, problems that require longer solutions are typically more complex and involve more reasoning steps. Comparing the `SFT(Difficult)` and `SFT(Difficult)+GRPO` curves, we can see that SFT with too complex expert traces can even hurt the performance of SLM, which further hurts the later GRPO stage. Finally, from the `GRPO w/ Expert trace` curve, we see that adding the expert trace as an extra rollout during GRPO mitigates sparse rewards and shows noticeable performance improvement, but it is still worse than BREAD. We suspect this is because there is a distribution gap between small and large model's reasoning traces, which makes SLM imitation of large models hard, while BREAD can reduce the gap during learning by letting the SLM figure out the reasoning steps by itself more. See further discussion below on **"hard questions"** for elaboration. We also provide more results among different base models and datasets in Table 2 and Appendix B.

**BREAD reduces training time.** BREAD is also markedly more training-efficient. It can reach the accuracy of the best baseline, `SFT+GRPO`, in just 25% of the training steps (Figure 7a), translating to roughly a 75% reduction in total compute FLOPs (Figure 7b). We estimated the FLOPs based on the common cost evaluation method used in recent scaling-law studies [35, 16, 30], by counting a forward pass as $2ND$ and a backward pass as $4ND$. Here $N$ is the number of model parameters and $D$ is the total token count processed in that pass. For calculation details, see Appendix A.1. Note that we estimate $D$ as the average length of all expert traces in the training set, but according our measurements, BREAD's actual generation length is always shorter. Therefore for BREAD, its actual token count, and thus its FLOP cost, is even lower than the estimated values reported Figure 7b. We also discuss the potential of reducing the number of rollout needed to reduce training cost in Appendix B.2.

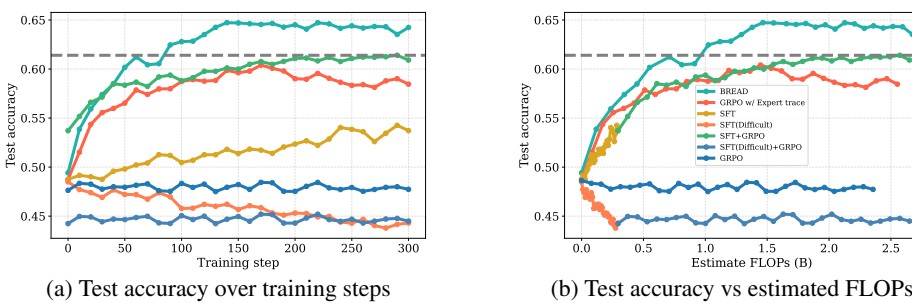

(a) Test accuracy over training steps      (b) Test accuracy vs estimated FLOPs

Figure 7: Test accuracy over training steps (left) / FLOPs (right). BREAD, which adaptively uses hints from expert traces during GRPO, significantly improves SLM reasoning ability compared to all baselines. The gray dashed line (max accuracy of the best baseline) demonstrates that BREAD can speed up the convergence speed by about $3\times$. Both figures share the same legend.

**BREAD succeeds on the hard questions, where other baselines fail.** To understand the gains of BREAD, we train and evaluate each method exclusively on very difficult problems. The goal is to investigate whether expert hints in BREAD can help SLMs learn new information particularly from these hard questions. We conduct the experiment on the NuminaMath-CoT dataset and the Qwen2.5-3B-Instruct as the base model. To build the hard dataset, we first run ordinary SFT and from the training dataset, we select the 500 questions for which three independent generations produce no correct trace (pass@$3 = 0$). These tasks are unsolved by the small model, and their expert traces

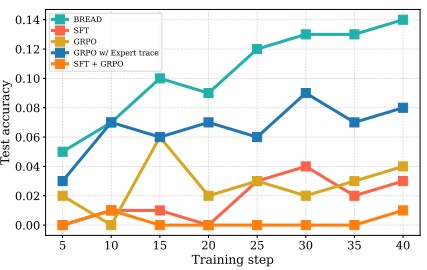 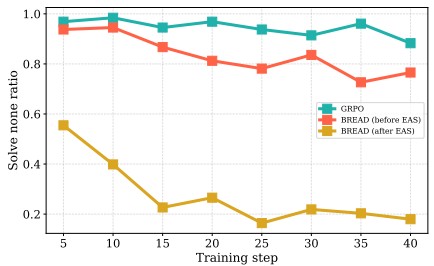

(a) Test accuracy over training steps      (b) Ratio of questions whose rollouts are all wrong

Figure 8: (a) Test accuracy on very hard questions over training steps. BREAD outperforms other baselines significantly while the traditional methods (SFT, GRPO, SFT+GRPO) learn little. (b) Proportion of test questions for which every rollout fails (solve-none ratio). In vanilla GRPO, the ratio stays persistently high, signalling that training stalls under sparse rewards. In BREAD, the solve-none ratio for regular rollout starts similarly high, but EAS injects expert traces and densifies the reward. This rate continuously decreases, confirming that the model is learning.

proved too complex for SFT to learn. The 500 samples were split into 80/20 train/test subsets. We then again used Qwen2.5-3B-Instruct as the base model and trained each method on this hard subset. The results are shown in Figure 8a. All baselines show little or no improvement on the test set after training on the hard questions. In contrast, BREAD achieves a clear upward performance with continued training, demonstrating that its partial expert guidance and branched rollout strategy can provide learnable information, even when standard SFT and vanilla GRPO fail. We argue that BREAD succeeds because it adaptively reduce problem difficulty and densifies the rewards. As shown in Figure 8b, BREAD sharply lowers the solve-none ratio, getting more informative samples and richer feedback. This enables BREAD to learn effectively even from very hard questions and complex reasoning traces.

**BREAD improves sample efficiency during training.** SLMs distilled via large-scale SFT can achieve strong reasoning capability, such as DeepSeek-R1-Distill [13]. However, the distillation pipeline is prohibitively expensive. For example, this model is trained with 800k samples from both reasoning and non-reasoning domains curated with DeepSeek-R1, containing 671 billion parameters. A single forward pass through such a huge model already costs more FLOPs than 25 RL training steps with 8 rollouts. The pipeline also needs expensive sample filtering and trace post-processing, like the heavyweight data-collection procedure in [24, 22]. What makes things worse is that, as illustrated in Table 1, each target model requires training with different samples, multiplying the cost.

In contrast, BREAD is far more sample-efficient, achieving comparable gains with a small fraction of the expert trace and without any heavyweight sample selection stage. To show this, we created two trace-budget–matched baselines, SFT(selected) and SFT(random). For fair comparison, we first record the expert traces actually requested by BREAD via Episode Anchor Search (EAS). On NuminaMath-CoT, this corresponded to 36.7% of samples, and on the easier Math [15] dataset the fraction falls to 19.1% (during 300 training steps of Qwen-2.5-3B-Instruct). We then supervised fine-tune base models with the same number of traces To create SFT(selected), we select the exact subset chosen by BREAD. To create SFT(random), we use an equally sized randomly picked subset. Finally, we created SFT(full), which uses all the expert traces, take a high cost. Each of the SFT(x) phases was followed by a GRPO phase. As the results in Table 2 show, BREAD outperforms nearly all the SFT(X)+GRPO baselines in terms of accuracy, and can even approach the performance of the expensive DeepSeek-R1-Distill model. Notably, while DeepSeek-R1-Distill gains from vast and diverse training data, BREAD achieves nearly comparable accuracy without requiring this data. We also observe that small models do not necessarily benefit from SFT with expert traces, as evidenced by the Qwen-2.5-1.5B-Instruct run on NuminaMath-CoT, where SFT(full)+GRPO has relatively low accuracy. Also, which samples are most useful for SFT is uncertain: on the MATH dataset, SFT(selected)+GRPO has higher accuracy than SFT(random)+GRPO with Qwen-3B-Instruct, but it is the opposite for Qwen-1.5B-Instruct.

## 5 Related Work

**Supervised fine-tuning (SFT) and Reinforcement Learning (RL).** Recently, there is a debate about whether SFT or RL can truly improve the reasoning ability of language models (LMs). With the emergence of [33, 13, 36], more and more reinforcement finetuned reasoning models demonstrate the importance of RL. in reasoning tasks. [6] prove that RL can enhance the reasoning ability of LMs

| Model | Dataset | GRPO | SFT(full)+GRPO | SFT(random)+GRPO | SFT(selected)+GRPO | BREAD | DeepSeek-Distill |
|-------|---------|------|----------------|------------------|--------------------|-------|------------------|
| Qwen-1.5B-Instruct | MATH | 0.590 | 0.774 | 0.712 | 0.706 | 0.788 | 0.818 |
| Qwen-3B-Instruct | MATH | 0.681 | 0.846 | 0.735 | 0.794 | 0.843 | / |
| Qwen-1.5B-Instruct | NuminaMath-CoT | 0.347 | 0.242 | 0.356 | 0.234 | 0.361 | 0.368 |
| Qwen-3B-Instruct | NuminaMath-CoT | 0.475 | 0.537 | 0.519 | 0.502 | 0.647 | / |

Table 2: BREAD outperforms other baselines and nearly reaches the accuracy of DeepSeek-R1-Distill, which has the benefit of vast training data. There is no DeepSeek-R1-Distill for Qwen-3B provided by [13], so its cells are left blank.

while SFT can only force LMs to memorize knowledge. However, [42] argues that the base model already has the reasoning ability while Reinforcement Learning with Verifiable Rewards (RLVR) barely increases the probability of correct reasoning trace. Furthermore, distillation works such as [13, 24] prove that SFT can help SLMs acquire reasoning capability comparable to the expert/teacher models. While the current popular pipeline to train a reasoning LM is SFT followed by RL, we argue the need for stronger integration of SFT and RL because, for hard questions, the expert solution might not be suitable for base models to learn while RL can struggle to discover even a single correct trace.

**Efficient RL and Reasoning.** While reasoning LMs become more powerful, computation is more demanding during the training and deployment procedure. Therefore, researchers recently paid more attention to efficient reasoning in both directions. Pivotal Token Search (PTS) [1] accelerates the Direct Preference Optimization (DPO) [27] training by identifying tokens in a language model generation that significantly impact the probability of success for the reasoning task. Following the memory efficiency but training time inefficiency introduced by GRPO [33, 13], many follow-up works improve the training convergence speed, including DAPO [41], CPPO [23], PODS [40] and DUMP [38]. [36, 43, 2] saves the token usage during inference by training with one or multiple levels of length penalty. Meanwhile, meta reinforcement fine-tuning (MRT) [26] makes inference token wasted less in meaningless reasoning steps by making the success rate steadily increase with the number of reasoning episodes. There are also orthogonal approaches to efficiency e.g. by forming cascades of small and large language models [5, 44, 14]. Compared with these previous works, we not only decrease the supervised signals during training by only introducing an expert solution when the current model cannot solve the current task with a probability relatively high enough, but guide the model with the expert hint to increase the RL training efficiency. which can provide a denser reward for speeding up the RL training procedure.

**Related classical RL methods.** DAgger [29] intermittently injects expert actions to correct agent behavior. Instead, BREAD adaptively inserts expert hints only when the agent fails and allows the model to complete the remaining of the reasoning trace, allowing for a natural curriculum. Go-Explore [8] trains an agent that can solve all Atari games by branching from intermediate states, while BREAD branches out from expert traces. Methods like Hindsight Experience Replay (HER) [3] and potential-based reward shaping (PBRS) [25] densifies learning signals under sparse rewards through goal relabeling or external shaping functions, but BREAD can achieve a similar effect through hint-based rollout initialization. Finally, while prior works like Kickstarting [32] and Expert Iteration [4] explore teacher-student transfer via full trajectories or alternating control, BREAD explicitly focuses on sparse expert intervention with minimal demonstrations. Some earlier work, such as Model-Based Policy Optimization (MBPO) [18], propose to automatically update the predictive model during policy training and only learn through short rollouts, but it does not include the hint from expert traces, which will lead to the improvement limitation in LLM post-training domain.

## 6   Discussion and Limitations

Our work introduces BREAD as a principled algorithm to densify the reward signal using branched rollouts from the expert trace. BREAD significantly enhances sample complexity, training time, and eventual accuracy over employing GRPO. It is also theoretically well motivated and allows the model to overcome fundamental bottlenecks of SFT+GRPO. BREAD has also a few limitations: Firstly, we focus on SLMs and assume that there is a strong expert/teacher model to provide high-quality traces. This assumption can be weakened as we can use BREAD whenever successful traces are obtained and use standard RL otherwise. Secondly, it is possible that SLM may fail to obtain a reward signal even from partial traces of the teacher. In the extreme scenario, the student model can't even conclude even if the full solution by the expert is presented as it can't follow the argument at all. This shortcoming could potentially be addressed by training or prompting the expert model to generate simpler more digestible traces. We leave these as future directions.

## Acknowledgements

This work is supported by the National Science Foundation grants CCF-2046816, CCF-2403075, CCF-2212426, the Office of Naval Research grant N000142412289, and an Adobe Data Science Research Award. The computational aspects of the research is generously supported by computational resources provided by the Amazon Research Award on Foundation Model Development.

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

**Algorithm 2 BREAD**: GRPO with Branch Rollouts and Expert Anchors (full version)

---

**Require:** Dataset $\mathcal{D}$, current policy $\pi_\theta$, sampling number $G$, a list of keywords for splitting episodes $[w_0, w_2, ..., w_J]$
1: **procedure** BREAD$(\mathcal{D}, \pi_\theta, G, [w_0, w_2, ..., w_J])$
2:     **for** step $= 1, 2, ..., N$ **do**
3:         Sample a batch $\mathcal{D}_b$ from $\mathcal{D}$
4:         Update the old policy model $\pi_{\text{old}} \leftarrow \pi_\theta$
5:         **for** each question and expert solution pair $(Q, S)$ in $\mathcal{D}_b$ **do**
6:             $[R_1, R_2, ..., R_G], \rho \leftarrow$ EAS$(Q, S, \pi_\theta, G, [w_0, w_2, ..., w_J])$ ▷ Episode Anchor Search (EAS)
7:             Add the $(Q, \rho), R_i$ to the buffer.
8:         **end for**
9:         For each $(Q, \rho), o_i$ in the buffer, compute $\hat{A}_{i,t}$ for the $t$-th token of $o_i$        ▷ Equation (2)
10:     **end for**
11:     **for** iteration $= 1, 2, ..., T$ **do**
12:         Update the policy model $\pi_\theta$ by maxmizing the BREAD objective        ▷ Equation (1)
13:     **end for**
14:     **return** $\pi_\theta$
15: **end procedure**
16: **procedure** EAS$(Q, S, \pi, G, [w_0, w_2, ..., w_J])$
17:     $[R_1, R_2, ..., R_G] \leftarrow \pi(Q)$        ▷ Sample $I$ responses for question $Q$ with current policy $\pi$
18:     $p_{\text{correct}} \leftarrow$ compute_correct_probability$([R_1, R_2, ..., R_G], S)$
19:                         ▷ compute correct probability based on responses and the gold solution
20:     **if** $0 < p_{\text{correct}}$ **then**
21:         **return** $[R_1, R_2, ..., R_G], NA$
22:     **else**
23:         $[e_0, e_1, ..., e_K] \leftarrow$ split_episodes$(S, [w_0, w_2, ..., w_J])$
24:         **return** BINARY_SEARCH_AND_GENERATE$(Q, S, \pi, G, [e_1, e_2, ..., e_K], 0, K)$
25:     **end if**
26: **end procedure**
27: **procedure** BINARY_SEARCH_AND_GENERATE$(Q, S, \pi, G, [e_1, e_2, ..., e_K], L, R)$
28:     $M = \frac{L+R}{2}$
29:     $[R_1, R_2, ..., R_G] \leftarrow \pi([Q, e_1, e_2, ..., e_M])$
30:     $p_{\text{correct}} \leftarrow$ compute_correct_probability$([R_1, R_2, ..., R_G], S)$
31:     **if** $0 < p_{\text{correct}} < 1$ **then**
32:         **return** $[R_1, R_2, ..., R_G], S_{1:M}$
33:     **else if** $p_{\text{correct}} = 0$ **then**
34:         $L = M, R = R, M = \frac{L+R}{2}$
35:         **return** binary_search_and_generate$(Q, S, \pi, G, [e_1, e_2, ..., e_K], L, R)$
36:     **else**
37:         $L = L, R = M, M = \frac{L+R}{2}$
38:         **return** binary_search_and_generate$(Q, S, \pi, G, [e_1, e_2, ..., e_K], L, R)$
39:     **end if**
40: **end procedure**

---

# A Experiment details

## A.1 FLOPs

We compute the estimate FLOPS following [35, 16, 30]. The supervised finetuning, which include one forward and one backward phase, people use a common approximation $6ND$ [16], and for inference, which include only one forward phase, people always use $2ND$ [30]. Here $N$ represents model parameters, $D$ is the total token count processed in that pass. So a forward phase takes $2ND$ while a backward phase take $4ND$. For estimation, we define the average length of a single question in one inference time as $D_{\text{sample}}$, so $D = D_{\text{sample}} \times n_{\text{rollout}}$ for RL and $D = D_{\text{sample}}$ for SFT. For estimation, we define the average length of a single question in one inference time as $D_{\text{sample}}$, so $D = D_{\text{sample}} \times n_{\text{rollout}}$ for RL and $D = D_{\text{sample}}$ for SFT. GRPO with eight rollouts, $n_{\text{rollout}} = 8$ including eight forward and eight backward phase needs $6 \times 8 \times ND_{\text{rollout}}$. BREAD will take more forward pass to do the binary search, so we use $6 * 8 * ND_{\text{rollout}} + 4ND_{\text{additional}}$. GRPO w/ Expert trace includes including eight forward and nine backward phase, so we use $6 \times 8 \times ND_{\text{rollout}} + 4 \times ND_{\text{rollout}}$. We estimate $D$ with the average length of all expert trace in the training dataset. We truly record the additional number of generation $D_{\text{additional}} = D_{\text{rollout}} \times n_{\text{additional\_rollout}}$. Notably, the generation length of BREAD is always shorter than expert trace and vanilla GRPO. So our BREAD can reduce even more computation resource compare to 75% shown in Figure 7b.

## A.2 Experiment Setting Details for BREAD

In addition to Section 4, we list additional details of our experiments here. During training, we set `max_prompt_length` $= 2048$ for MATH experiments and `max_prompt_length` $= 4096$ for NuminaMath-CoT experiments. For both training and evaluation, we set `max_response_length` $= 4096$ for MATH experiments and `max_response_length` $= 8192$ for NuminaMath-CoT experiments. During training, we set the number of rollouts as 8. We used a low-variance KL divergence and set the coefficient for KL divergence as 0.001. We set the temperature at 0.6 for both training and evaluation.

For the implementation details of the episode splitting for the expert trace during training, we split the expert trace with sentences separated by ''. '' or ''\n'' for MATH and paragraphs separated by ''\n\n'' for NuminaMath-CoT. Note that the expert traces can be either human provided solutions or correct solutions provided by larger expert models, and the splitting method here can also be changed to split according to a specific keyword list. After splitting, we aggregate the traces into 10 episodes evenly for all expert traces. The reason why we implement this way is that we want to make sure that the number of episodes of the expert traces is not too large, which can guarantee that the EAS step does not take too much time.

During training, assume we have a question $Q$ and an expert hint $\rho$ ( or without $\rho$ during inference), the template of the prompt is as follows:

```
{'content':  '<|im_start|>system\nYou are a helpful assistant.  You
first thinks about the reasoning process in the mind and then provides
the user with the answer.<|im_end|>\n<|im_start|>user\n{Q, ρ} Show your
work in <think> <\think> tags.  And return the final answer within
\\boxed{}.<|im_end|>\n<|im_start|>assistant\nLet me solve this step by
step.\n<think>', 'role':  'user'}
```

For the hardware requirements, all of experiments are done with 8 L40S 40GB GPUs except the training starting from Qwen2.5-3B-Instruct as the base model, which requires 8 80G H100 GPUs.

# B Additional Experiments

## B.1 SFT training saturates and does not help after a while

Supplement to Figure 7a. As shown in Figure 9, training SFT for more iterations does not further improve test accuracy. In other words, we already use the model that SFT can achieve. For a fair comparison, we therefore use the 300-step checkpoint as the starting point for SFT+GRPO.

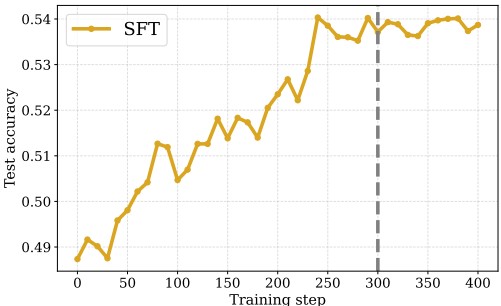

Figure 9: Test accuracy of SFT over training steps. The accuracy doesn't continuously increase after the number of training step we use.

## B.2 BREAD can reduce the number of rollouts needed

Another straightforward way to reduce training cost is to reduce the number of rollouts, since FLOPs scale linearly with that count. However, for vanilla GRPO, fewer rollouts lead to lower accuracy, whereas BREAD maintains its performance even as the number of rollouts decreases. The result is shown in Figure 10. As we have shown in the main body, when the task is so difficult that the base policy rarely produces correct traces, vanilla GRPO fails. To study rollout efficiency under conditions where GRPO can learn, we moved to the MATH dataset with a Qwen-2.5-3B-Instruct base model. With eight rollouts, GRPO does increase accuracy. However, performance will decrease if the rollout budget is reduced from 8 to 5. In contrast, BREAD reaches the same accuracy with just five rollouts, cutting training FLOPs while preserving performance.

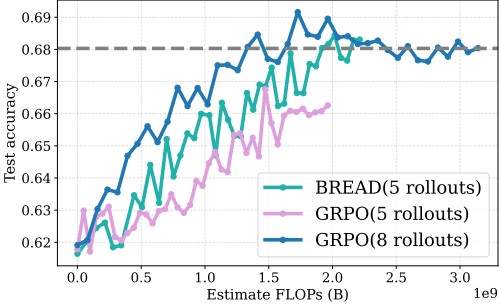

Figure 10: Test accuracy over training steps with different number of rollouts. The gray dashed line shows the final accuracy of vanilla GRPO with 8 rollouts. The total training step is 500.

## B.3 Episode details

As described in Appendix A.2, we plot the distribution of the number of steps for our two datasets before episode aggregation. As shown in Figure 11, we can see that that most expert traces contain less than 20 steps, while few of them contain much more steps, which may slow down the training sif there ised if there is no episode aggregation.

## B.4 BREAD can reduce the number of rollouts needed

We also conduct experiments with LLaMA3.2-3b-Instruct model. The results are shown in Table 3. These experiments support that the method generalizes beyond the Qwen family and is not model-specific, and the effectiveness is more significant when the task is harder.

## B.5 BREAD compare to other methods.

Regarding recent works, we would like to highlight that our focus is on challenging tasks where existing methods like DAPO [41] are not specifically designed to perform well. So, these tasks are

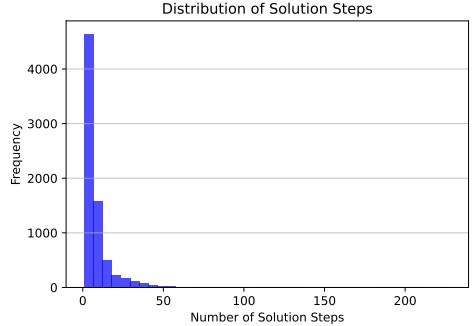
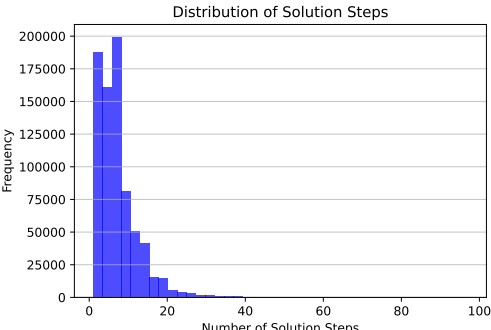

(a) Distribution plot of MATH solution step numbers.

(b) Distribution plot of NuminaMath-CoT solution step numbers.

Figure 11: Distribution plot of MATH and NuminaMath-CoT solution step numbers

| Method | MATH (Full) | MATH (Hard Subset) |
|---|---|---|
| GRPO | 43.6% | 0.0% |
| SFT+GRPO | 45.9% | 3.0% |
| BREAD | 48.2% | 6.0% |

Table 3: Performance of LLaMA3.2-3B-Instruct model on the full MATH dataset and its hard subset (constructed following Figure 8).

typically not included in the evaluation benchmarks of DAPO and similar methods. For example we train and evaluate DAPO on the same difficult dataset used in Figure 8, under the same settings. We observed that DAPO shows little improvement. This aligns with its design: DAPO's dynamic sampling strategy tends to filter out hard questions rather than try to solve them, making it less useful for the types of tasks we target with BREAD. The results are shown in Table 4.

| Method | Accuracy |
|---|---|
| GRPO | 4% |
| SFT+GRPO | 1% |
| BREAD | 14% |
| DAPO | 5% |

Table 4: BREAD compare to DAPO.

## C `GRPO w/ Expert Trace` **Details and its Connection with RL with SFT Loss**

The baseline `GRPO w/ Expert Trace` (GRPO-ET) generally follows the same procedure as the standard GRPO. Instead, it enforces one of the $G$ rollouts as the expert trace. Therefore, the objective function is

$$
\mathcal{J}_{\text{GRPO-ET}}(\theta) = \mathbb{E}_{(q,S,a)\sim\mathcal{D},\{o_i\}_{i=1}^{G-1}\sim\pi_{\text{old}}(\cdot|q)}
$$
$$
\left[\frac{1}{G}\left(\sum_{i=1}^{G-1}\frac{1}{|o_i|}\sum_{t=1}^{|o_i|}\left(\min\left(r_{i,t}(\theta)\hat{A}_{i,t},\text{clip}\left(r_{i,t}(\theta),1-\varepsilon,1+\varepsilon\right)\hat{A}_{i,t}\right)-\beta D_{\text{KL}}(\pi_\theta||\pi_{\text{ref}})\right)\right)\right]
$$
$$
+\frac{1}{G|o_S|}\sum_{t=1}^{|o_S|}\left(\min\left(r_{i,t}(\theta)\hat{A}_{S,t},\text{clip}\left(r_{i,t}(\theta),1-\varepsilon,1+\varepsilon\right)\hat{A}_{S,t}\right)-\beta D_{\text{KL}}(\pi_\theta||\pi_{\text{ref}})\right) \quad (3)
$$

where

$$
r_{i,t}(\theta) = \frac{\pi_\theta(o_{i,t}|q,o_{i<t})}{\pi_{\text{old}}(o_{i,t}|q,o_{i<t})}, \quad \hat{A}_{i,t} = \frac{R_i - \text{mean}(\{R_i\}_{i=1}^{G-1},R_S)}{\text{std}(\{R_i\}_{i=1}^{G-1},R_S)}, \quad \hat{A}_{S,t} = \frac{R_S - \text{mean}(\{R_i\}_{i=1}^{G-1},R_S)}{\text{std}(\{R_i\}_{i=1}^{G-1},R_S)}
$$
$$
(4)
$$

All notations in Equations (3) and (4) are the same as [13], while $S$ represents the expert trace, and all variables whose subscript contains $S$ represent the corresponding variables.

Here, we can see a clear connection between `GRPO w/ Expert Trace` and GRPO (rollout number is $G - 1$) with an SFT loss. Suppose that we want to deploy GRPO while adding the SFT entropy loss to the standard GRPO loss, the objective function is

$$\mathcal{J}_{\text{GRPO\_SFT}} = \mathbb{E}_{(q,S,a) \sim \mathcal{D}, \{o_i\}_{i=1}^{G-1} \sim \pi_{\text{old}}(\cdot|q)}$$

$$\left[ \frac{1}{G} \left( \sum_{i=1}^{G-1} \frac{1}{|o_i|} \sum_{t=1}^{|o_i|} \left( \min\left( r_{i,t}(\theta)\hat{A}_{i,t}, \text{clip}\left(r_{i,t}(\theta), 1-\varepsilon, 1+\varepsilon\right)\hat{A}_{i,t}\right) - \beta D_{\text{KL}}(\pi_\theta||\pi_{\text{ref}}) \right) \right) \right]$$

$$+ \sum_{t=1}^{|o_S|} \log \pi_\theta(S_t \mid q, S_{<t}) \tag{5}$$

where

$$r_{i,t}(\theta) = \frac{\pi_\theta(o_{i,t}|q, o_{i<t})}{\pi_{\text{old}}(o_{i,t}|q, o_{i<t})}, \quad \hat{A}_{i,t} = \frac{R_i - \text{mean}(\{R_i\}_{i=1}^{G-1}, R_S)}{\text{std}(\{R_i\}_{i=1}^{G-1}, R_S)} \tag{6}$$

There are 3 main difference between Equation (3) and Equation (5):

1. A coefficient for the expert trace loss $\frac{1}{G|o_S|}$

2. A KL divergence term $-\beta D_{\text{KL}}(\pi_\theta||\pi_{\text{ref}})$

3. An Advantage term $\min\left( r_{i,t}(\theta)\hat{A}_{S,t}, \text{clip}\left(r_{i,t}(\theta), 1-\varepsilon, 1+\varepsilon\right)\hat{A}_{S,t}\right)$

Suppose that $r_{i,t} \leqslant 1 + \varepsilon$, because $\hat{A}_{S,t} \geqslant 0$, $\min\left( r_{i,t}(\theta)\hat{A}_{S,t}, \text{clip}\left(r_{i,t}(\theta), 1-\varepsilon, 1+\varepsilon\right)\hat{A}_{S,t}\right) = r_{i,t}(\theta)\hat{A}_{S,t}$, which is exactly the entropy loss with a coefficient $\hat{A}_{S,t}$ if we replace $\pi_{\text{old}}(o_{i,t}|q, o_{i<t})$ in $r_{i,t}(\theta)$ with the one hot embedding of the expert trace tokens. Therefore, `GRPO w/ Expert Trace` can not only have better training consistency, but can also assign different credits according to the token probability ratio between the old and the current policy.

---

**Algorithm 3 GRPO w/ Expert Trace** (GRPO-ET)

---

**Require:** Dataset $\mathcal{D}$, current policy $\pi_\theta$, sampling number $G$
1: **procedure** GRPO_with_Expert_Trace($\mathcal{D}, \pi_\theta, G$)
2:     **for** step $= 1, 2, ..., N$ **do**
3:         Sample a batch $\mathcal{D}_b$ from $\mathcal{D}$
4:         Update the old policy model $\pi_{\text{old}} \leftarrow \pi_\theta$
5:         Sample $G - 1$ outputs $\{o_i\}_{i=1}^{G-1} \sim \pi_{\text{old}}(\cdot|Q)$ for each question $Q \in \mathcal{D}_b$
6:         Combine the expert trace with the sampled outputs to construct $\{\{o_i\}_{i=1}^{G-1}, S\}$
7:         Compute rewards $\{\{r_i\}_{i=1}^{G-1}, r_S\}$ for each sampled output $o_i$ and expert trace $S$
8:                                           $\triangleright$ $r_S$ is generally 1 because of solution correctness
9:         For each $o_i$ and expert trace $S$. compute $\hat{A}_{i,t}$ for the $t$-th token of $o_i$ and $S$.     $\triangleright$ Equation (4)
10:         **for** iteration $= 1, ..., T$ **do**
11:             Update the policy model $\pi_\theta$ by maximizing the GRPO-ET objective     $\triangleright$ Equation (3)
12:         **end for**
13:     **end for**
14:     **return** $\pi_\theta$
15: **end procedure**

---

# D   Supporting Material for Section 2.1: Proofs and Experimental Details

## D.1   Experimental Details

Recall that we consider a Markov chain with $K + 1$ states (indexed by $0, 1, 2, \cdots, K$) and a $(K + 1) \times (K + 1)$ transition matrix. We assume that, for the expert/large model, all $(i \rightarrow j)$ transitions are learnable in the transition matrix. However, for a small/student model, not all $(i \rightarrow j)$ transitions

are learnable and we denote the learnable state transitions by $\mathcal{P} \subset [K+1] \times [K+1]$ where $[K] = \{0, 1, 2, \ldots, K\}$.

We first introduce the following definition of pretrained small model utilized in our experiments.

**Definition 1 (Pretrained Small Model)** *A pretrained small model is defined as a Markov model* $\mathcal{M} = (\mathcal{S}, \boldsymbol{P})$, *where* $\mathcal{S} = \{0, 1, 2, \ldots, K\}$ *is the state space and* $\boldsymbol{P} \in \mathbb{R}^{(K+1) \times (K+1)}$ *is the transition matrix. The model satisfies the following properties:*

- *Let* $d \geqslant 1$ *denote the maximum allowed jump between states. The set of learnable state transitions* $\mathcal{P}$ *satisfies:*

$$(i, j) \in \mathcal{P} \quad \text{if and only if} \quad |i - j| \leqslant d.$$

- *For some* $\epsilon \ll 1/d$, *the transition probabilities satisfy:*

$$\boldsymbol{P}_{ij} := \mathbb{P}(i \rightarrow j) = \begin{cases} 1 - \Theta(d\epsilon), & \text{if } i = j, \\ \Theta(\epsilon), & \text{if } (i, j) \in \mathcal{P} \text{ and } i \neq j, \\ 0, & \text{if } (i, j) \notin \mathcal{P}. \end{cases}$$

In the following, we describe the BREAD algorithm for the Markov model. Our algorithm is in line with the theory presented in Section 2.2 and incrementally learns a solution by optimizing the local transition dynamics starting from the right end of the expert trace.

---

**Algorithm 4 BREAD_Markov**

---

**Require:** An SFT trajectory $[\alpha_0, \cdots, \alpha_T]$, initial small model $\mathcal{M}([K], \boldsymbol{P})$, reward threshold $r_{\text{thred}}$, maximal trajectory length $T_{\max}$
1: **for** episode $t = T, T-1, \ldots, 1$ **do**
2:      $r \leftarrow 0$
3:      **while** $r < r_{\text{thred}}$ **do**
4:          Sample $N$ trajectories from $\mathcal{M}$ starting from state $\alpha_t$ with maximal length $T_{\max} - t$
5:          Update the current reward: $r \leftarrow r_{\text{new}}$
6:          Update the transition matrix: $\boldsymbol{P} \leftarrow \boldsymbol{P}_{\text{new}}$
7:      **end while**
8: **end for**

---

**Experimental settings for Figure 3:** We conduct experiments by fine-tuning a Markov chain model (cf. Definition 1) using three different methods: SFT, GRPO, and BREAD. To introduce additional randomness in our experiments, for each state $i \in [K]$, we randomly select one of its connected states (e.g., among $i - d, \ldots, i, i + d$ states) and assign it the highest transition probability of $1 - \Theta(\epsilon)$, instead of always assigning the highest probability to $\mathbb{P}(i \rightarrow i)$. Due to symmetricity, this is not expected to change the fundamental behavior of the algorithms but provides more variability. In both Figs. 3a and 3b, $d = 2, T_{\max} = 2 \cdot K$ and SFT is infeasible with expert jump size of 3 as we set $T = K/3$. In Fig. 3a, we fix $K = 30$ and vary $\epsilon \in \{0.01, 0.025, 0.05\}$. In contrast, in Fig. 3b, the $\epsilon = 0.05$ is remained unchanged and the number of states varies in $K \in \{30, 50, 100\}$. Each experiments is trained for 10000 iterations with 1000 trajectories sampled with each iteration.

## D.2 Proofs in Section 2.2

### D.2.1 Proof of Lemma 1

Since the expert model always jumps two or more whereas the student is restricted to a jump size of at most one, the SFT phase (specifically maximizing log-likehood using expert trace) will have no impact on the student model. This is because the supervised loss, induced by the negative log-likelihood, is not optimizable beyond infinity due to 0 transition probabilities assigned between $x_t \rightarrow x_{t+1}$. To conclude the proof, we show that, no trajectory sampled during the RL phase will receive a reward with high probability. Since the SFT phase has no impact on the base student model, the student still follows a symmetric random walk. Let $(x_i^s)_{i=1}^L$ denote this random walk which is a sum of IID Rademacher variables. The classical concentration bounds on the hitting-time of random walks state [7, 10]

$$\mathbb{P}(\exists \, x_i^s = K \text{ for } 1 \leqslant i \leqslant L) \leqslant 2e^{-K^2/2L}.$$

Thus, with access to $N \leqslant e^{K^2/4L}$ samples, applying union bound, we guarantee that no trace generated by the student model receives a reward.

### D.2.2 Proof of Theorem 1

We first recall the following lemma on random walk hitting times.

**Lemma 2 (Lower bound on random walk hitting time, [10])** *If $X_i$ is a symmetric random walk initialized at $0$, we have that $\mathbb{P}(\max_{1 \leqslant i \leqslant L} X_i \geqslant K) \geqslant 1 - 0.8K/\sqrt{L}$.*

To proceed, the proof will be done inductively. Suppose for all rounds $i < \tau$, the followings hold:

- Each round requires at most $t$ traces to achieve success with probability at least $1 - e^{-t}$.
- All sub-traces generated by the small model at round $i$ (starting from the expert's $x_{T-i}$) that are stitched to $S_{i-1}$ (trace constructed so far) is at most length $L/T$.

To complete the induction, we consider the new round $\tau$ and prove the two statements above. At round $\tau$, we start from State $x_{T-\tau}$ and aim to reach a State within $S_{\tau-1}$. This process will follow a symmetric random walk initialized at $x_{T-\tau}$. By design, $x_{T-\tau+1}$ is within $S_{\tau-1}$. As a result, the trajectory length of the subtask is upper bounded by the time it takes for the random walk to reach from $x_{T-\tau}$ to $x_{T-\tau+1}$. Consequently, the total trajectory length is upper bounded by the sum over all subtasks. Since each subtask is independent and identically distributed, we focus on analyzing a single subtask with maximal trace length $L/T$. Recall from the assumptions in Section 2.2 that $|x_{T-\tau} - x_{T-\tau+1}| \leqslant c \cdot K/T$. To this aim, we apply Lemma 2 which ensures that, probability of hitting within $L/T$ trace length is at least $1 - 0.8\frac{c \cdot K/T}{\sqrt{L/T}} = 1 - 0.8\frac{c \cdot K}{\sqrt{LT}}$. Since $L \geqslant 5c^2K^2/T$, probability of failure for each trace is $\leqslant 1/e$. Over $t$ IID traces, probability of success at any round $i$ is at least $1 - e^{-t}$. Aggregating these events, we find that the probability of success over $T$ rounds is at least $1 - Te^{-t}$ given $t$ BREAD rollouts at each round.

