# OpenReview forum: "BREAD: Branched Rollouts from Expert Anchors Bridge SFT & RL for Reasoning"
_NeurIPS.cc/2025/Conference — NeurIPS 2025 poster_

### Official Review · Reviewer_KHRX · 2025-06-29

**Clarity:** 3
**Significance:** 3
**Originality:** 3
**Rating:** 5
**Confidence:** 4

**Summary:**

The paper explores reasoning with (relatively) small language models. The authors propose BREAD, a method that combines RL and SFT via a simple strategy: if the policy model cannot solve the given question, it is given a hint via a partial expert trajectory on this prompt. More precisely, BREAD does a binary search to figure out the length of the partial trajectory that leads to the student model to achieve pass rates within a pre-defined range. The authors show that this simple procedure outperforms combinations of supervised finetuning (SFT) and reinforcement learning (GRPO).

**Questions:**

**Q1.** When you do policy updates, do you train on all tokens, including the expert tokens? Or only the completion tokens generated by the student? Is it true that the method is no longer a policy gradient method, as the trajectories are off-policy?

**Q2.** Do I understand correctly that all plots in Fig 6 and Fig 7 are test pass rates, where the test set is separate from the train set, and you are not training on the test set?

**Q3.** Same question for Table 2: for these results, do you use any transcripts on MATH and Numina? What are the questions that you distill on?

**Q4.** Do I understand correctly that in your SFT experiments you use a much lower amount of SFT compared to the industry methods such as DeepSeek-Distill? What is the size of the "full" dataset in your experiments?

**Ethical Concerns:**

["NO or VERY MINOR ethics concerns only"]

**Final Justification:**

After reading the rebuttal and the other reviews, I maintain my score and recommend accepting the paper.

**Limitations:**

Limitations adequately addressed.

**Paper Formatting Concerns:**

No concerns.

**Quality:**

3

**Strengths And Weaknesses:**

# Strengths

**S1.** The paper is clear and easy to follow.

**S2.** The proposed method is very simple and makes intuitive sense. It is easy to implement, and it has a very clear intuition. It can probably also be generalized to other kinds of "hints" beyond partial trajectories.

**S3.** Section 3.1 presents a toy model where the improvement from BREAD can be proved theoretically.

**S4.** The empirical evaluation shows consistent improvements relative to other combinations of SFT and GRPO.

# Weaknesses

**W1.** If I understand correctly, the policy is updated by training on all the tokens, including the expert tokens which were not generated by the policy. If this is correct, I believe that the policy update is no longer a policy gradient update, since the trajectories can be significantly off-policy.

**W2 (minor).** Please say explicitly in the Policy Update section of Section 3 that the update (1) is the standard GRPO update. Otherwise, it is not clear where this objective comes from.

**W3.** The presentation often does not provide all the details of the results. For example, in Section 3.2 RL-only paragraph, it is unclear what model is used, what dataset, etc. Without these details, the information on how many problems are unsolved in each batch seems pointless. In the SFT-only section, it is stated that the SFT is done on "1000 difficult questions". What are those questions? Also, it is unclear what are S1K and S1K-1.1.

**W4.** Similarly, it is unclear what exactly is shown in Figure 5b. It is said that RL is using 40% hints, but then there is a full range of values for the ""Hint ratio" in the plot, and there is an improvement for 0% hint ratio. Please clarify what exactly is shown.

**W5.** In an ideal world, it would be great to see a comparison with GRPO and possibly newer methods like DAPO [1] in a standardized setting such that would be possible to directly compare to existing results in the literature (e.g. Qwen2.5-32B on AIME). I realize that this may be computationally challenging, and possibly 32B is already not a "small" LM. I note that the authors provide a comparison to DeepSeek-Distill in Table 2.

[1] DAPO: An Open-Source LLM Reinforcement Learning System at Scale

---

> ### Author Rebuttal · Authors · 2025-07-31
>
> Thank you for your kind and constructive feedback. We’re pleased that you found our approach valuable and have addressed your questions and concerns point by point below. We would be grateful to address further questions during the discussion period.
>
> **W1: Does BREAD introduce off-policy learning issues?**
>
> The expert traces are used to guide the trajectory generation, not to replace or bypass the model’s policy. In detail, BREAD samples rollouts by interpolating between the expert traces and the self-generated traces to adapt difficulty. However, the policy gradient update is always computed using the log-probabilities of the model’s own trace(online part). This ensures that the learning signals remain on-policy, even though the rollout paths may be partially guided by the expert. Therefore, BREAD benefits from expert traces without breaking the assumptions of policy gradient methods.
>
> **W2: Please say explicitly in the Policy Update section of Section 3 that the update (1) is the standard GRPO update.**
>
> A: Thank you for the suggestion. We will clarify in the next version.
>
> **W3,4: The presentation often does not provide all the details of the results.**
>
> A: Thank you for the suggestion. We will make this clearer in the next version. For the details of results in Section 3.2, the information is provided in the caption of Table 1: we use Qwen2.5-1.5B-Instruct and Qwen2.5-3B-Instruct as our base SLMs, and the S1K and S1K-1.1 datasets are generated following the procedure in [2]. Additional experimental details can be found in Appendix A.
>
> The hint ratio in the x-axis in Figure 5b refers to the proportion of hint tokens appended to the query during **inference**, which is distinct from the 40% hint ratio (orange squares) used during RL **training**. A higher hint ratio means more guidance for the model. Currently, this setting is explained in Lines 180–185 and 198–201. Our results show that conditioning RL training on partially hinted questions not only improves performance on hinted evaluation queries (e.g., those with 40% hints), but also leads to improved accuracy on fully unhinted queries.  This highlights the model’s ability to generalize from partial short traces to full long traces.
>
> **W5: Compare to other methods in a standardized setting.**
>
> A:  Thank you for the suggestion. We agree that standardized comparisons are valuable. In our work, we include vanilla GRPO and DeepSeek-Distill as baselines to enable such comparisons. Regarding recent works, we would like to highlight that our focus is on challenging tasks and SLMs where existing methods like DAPO[1] are not specifically designed to perform well. So, these settings are typically not included in the evaluation benchmarks of DAPO and similar methods. For example, DAPO use Qwen2.5-32B instead of smaller models on those challenging tasks,
>
> For example, we train and evaluate DAPO on the same difficult dataset used in Figure 7, under the same settings. We observed that DAPO shows little improvement. This aligns with its design: DAPO’s dynamic sampling strategy tends to filter out hard questions rather than try to solve them, making it less useful for the types of tasks we target with BREAD.
>
> |Method|Acc|
> |-|-|
> GRPO|4%|
> SFT+GRPO|1%|
> BREAD|14%|
> DAPO|5% |
>
> **Q1,2: Are we training on the test set?**
>
> A: No, we do not train on the test set. In Figure 6, we evaluate using the official test subset provided by the Math/Numina-CoT benchmark. In Figure 7, we create an 80/20 train/test split from the dataset to ensure a clear separation between training and evaluation data. And in Table 2, we train on the training set and evaluate using the test subset provided by the Math/Numina-CoT benchmark.
>
> **Q3: Training data for SFT**
>
> A: Yes, we use a much lower amount of data. For SFT(full), we use the training subset of MATH(about 7500 samples)/NuminaMath-CoT to do SFT. In contrast, DeepSeek-Distill uses a much larger dataset. They use approximately 800k samples, including both reasoning and non-reasoning samples.
>
> [1] DAPO: An Open-Source LLM Reinforcement Learning System at Scale
>
> [2] Niklas Muennighoff, Zitong Yang, Weijia Shi, Xiang Lisa Li, Li Fei-Fei, Hannaneh Hajishirzi, Luke Zettlemoyer, Percy Liang, Emmanuel Candès, and Tatsunori Hashimoto. s1: Simple test-time scaling. arXiv preprint arXiv:2501.19393, 2025.

---

> > ### Comment · Reviewer_KHRX · 2025-08-05
> > **Response to the authors**
> >
> > Dear authors, thank you for the response, and the additional results. I am happy to maintain my score.
> >
> > I have read the other reviews, and I agree with some of the concerns raised:
> > - It would be great to include results on models other than Qwen, given known issues with this family of models. The authors added some limited results on Llama in the response.
> > - The authors should cite the related work mentioned by the other reviewers.

---

> ### Author Response · Authors · 2025-08-05
>
> We thank the reviewer for their continued support and thoughtful comments.
>
> Regarding the use of Qwen models, we have conducted additional experiments on the LLaMA3.2-3b-Instruct model. The results are consistent with Qwen experiments. For the final manuscript, we will incorporate at least one more model family such as Phi3 or Phi4 models from Microsoft. To make sure our comparison is fair, we follow the LLaMA3 technical report, which uses a 0-shot CoT prompt. Specifically, because the technical report does not provide detailed prompt, we use a general and intuitive 0-shot CoT prompt and provide it here to make sure our result is reproducible:
> ```
> messages = [
>   {“role”: “system”, “content”: “You are a helpful assistant that reasons step by step to solve complex problems.“},
>   {“role”: “user”, “content”: QUERY + “Let’s think step by step.“}
> ]
> ```
> The result are for the LLaMA3.2-3b-Instruct model and are as follows:
>
> (1) On the full MATH dataset:
>
> |GRPO|SFT+GRPO|BREAD|
> |-|-|-|
> |43.6%|45.9%|48.2%|
>
> (2) We also identify a hard subset of MATH dataset using similar methodology as described in the paper (Figure 7). Training over this dataset yields:
>
> |GRPO|SFT+GRPO|BREAD|
> |-|-|-|
> |0.0%|3.0%|6.0%|
>
> These experiments support that the method generalizes beyond the Qwen family and is not model-specific, and the effectiveness is more significant when the task is harder. We will also conduct and incorporate more experiments for the final version.
>
> Finally, we will incorporate and provide a discussion of the additional related work cited by other reviewers in the final version. Thank you again for the helpful feedback.

---

### Official Review · Reviewer_iFM6 · 2025-07-01

**Clarity:** 4
**Significance:** 4
**Originality:** 4
**Rating:** 5
**Confidence:** 4

**Summary:**

This paper introduces Branched Rollouts from Expert Anchors, a novel approach that combines supervised fine-tuning and reinforcement learning to enhance reasoning in small language models. The authors identify the limitations of traditional SFT + RL training, particularly for SLMs struggling to learn from complex expert traces and sparse rewards. BREAD addresses these issues by incorporating partial expert guidance and branched rollouts, allowing for more efficient learning. The method reduces the need for extensive expert traces, accelerates training, and improves model performance, especially on challenging tasks.

**Questions:**

1. Have the authors tested whether models with 7B parameters or larger, when using the same SFT and RL methods as described in this paper to improve model COT capabilities, experience the issues of insufficient RL rewards or performance degradation caused by the SFT-only method mentioned in the paper?

2. Does the length of the COT reasoning path affect the training algorithm proposed in this paper? Typically, how many reasoning steps are involved in the expert traces constructed in this paper?

**Ethical Concerns:**

["NO or VERY MINOR ethics concerns only"]

**Final Justification:**

After considering the rebuttal and discussions, I believe the authors have made a strong effort to address the key concerns, and my overall impression of the paper has improved. I will keep the score.

**Limitations:**

yes

**Quality:**

3

**Strengths And Weaknesses:**

Strengths:
1. The BREAD framework effectively bridges the gap between supervised fine-tuning and reinforcement learning , addressing the issues of sparse rewards and weak initialization in small language models , ensuring more efficient learning.
2. BREAD demonstrates substantial improvements in training efficiency, reducing the number of required rollouts and thus lowering computational costs while achieving higher accuracy than traditional GRPO and SFT methods.
3. BREAD outperforms existing methods, including GRPO and SFT+RL, on challenging tasks, especially those that are unsolvable using standard SFT+RL, by providing expert guidance and adaptive curriculum learning.
4. The paper offers a thorough theoretical analysis, formalizing the limitations of SFT+RL and justifying the design of BREAD, supported by mathematical insights into how expert traces can enhance model performance.
5. Experimental results consistently show that BREAD not only achieves superior accuracy but also accelerates training time.

Weaknesses:

1. Section 3.2 on the Deficiencies of SFT-only: a) The authors have not provided sufficient evidence to demonstrate that the performance degradation observed with SFT-only is due to the model's insufficient learning capability rather than biases present in the SFT data. b) The authors have concluded that SFT-only leads to performance degradation based on a limited set of SFT datasets (only two) and a narrow range of SFT methods (two methods). I believe these experiments are insufficient to robustly prove the claim that SFT-only causes performance degradation.

2. Bias in the Datasets Used in Table 2: The datasets used in Table 2 are biased, as they are all mathematics-related, which may not adequately demonstrate the generalizability of the proposed method. Validating the method on other reasoning datasets would provide a better assessment of its broader applicability.

---

> ### Author Rebuttal · Authors · 2025-07-31
>
> We appreciate your positive evaluation of the manuscript and your thoughtful suggestions. Thank you for acknowledging our approach—please find our responses to your questions and concerns below. We would be grateful to address further questions during the discussion period.
>
> **W1: More evidence to conclude that performance degradation with SFT-only arises from model limitations.**
>
> A: In addition to empirical results on real datasets, we provide theoretical insights to illustrate the limitations of SFT as presented in Section 3.1. It is known that an L times deeper LLM can internally simulate L chain-of-thought steps of a smaller LLM [1]. So the expert model can generate a dense reasoning trace which necessitates an L times longer simplified trace for the student model to digest via SFT. Translating this to a toy student-expert model over Markov chains, we show how the student model cannot benefit at all from the expert model’s trace. As for more empirical evidence, some recent papers also have similar observations [2,3].
>
> We also would like to clarify that our intent in Section 3.2 is not to claim that SFT-only universally leads to performance degradation. Rather, we aim to show that the limitations of SFT can cause SFT + RL strategies to fail, particularly in complex reasoning tasks involving long traces. We highlight a specific failure mode that can arise even when using widely used SFT datasets, and our results serve to motivate the need for the proposed method BREAD. We also acknowledge the reviewer's concern and will clarify that the *strict degradation* SFT in Table 1 may stem from the distributional difference between S1K and MATH/GPQA datasets. Note that, we pick the s1k dataset because the s1 paper [8] shows that SFT with s1k achieves very strong results on the 32B model. So it is naturally surprising that our experiments demonstrate that the same training data can in fact hurt the performance of small models. We speculate that this is because the s1k dataset is too hard for small models (besides the distribution shift) but more digestible for large models. Furthermore, NuminaMath evals on Table 2 indicate that SFT can hurt performance even on the same distribution. We will carefully clarify these points in the final revision.
>
> **W2: All datasets are all mathematics-related.**
>
> A: Thank you for the thoughtful feedback. Mathematical reasoning tasks are widely used in prior works [3-9] as a standard benchmark for evaluating reasoning capabilities. Also, these tasks strike a critical balance, they are sufficiently challenging, yet solvable for SLMs with the correct training strategy. This makes them particularly well-suited for studying GRPO variants like BREAD.
>
> Additionally, the availability of reliable expert traces in this domain allows us to conduct meaningful and controlled experiments, which may not be feasible in other domains.
>
> Therefore, we believe that math benchmarks serve as a strong and appropriate testbed for analyzing the effectiveness of our method. But we really appreciate the reviewer’s concern and would welcome suggestions on alternative reasoning datasets. We are open to expanding our evaluation in future versions.
>
> **Q1: Did you observe sparse rewards in RL or performance degradation from SFT with larger models?**
>
> A: Yes. The spare reward issue can happen even with Qwen2.5-32B-Instruct, it achieves only 10% accuracy on AIME, which improves to 37% after fine-tuning on the S1K dataset. However, even at 37% accuracy, reward signals in RL remain sparse(19.5% of samples all 8 rollouts fail). BREAD targets the fundamental limitations of SFT and RL that are not constrained by the model size. Although these problems are more noticeable in small models, high-capacity models can also struggle with similar difficulties, particularly when dealing with complex reasoning tasks where high-quality traces are hard to generate and the RL reward is sparse due to low probability of success.
>
> **Q2: How many reasoning steps are involved in the expert traces constructed in this paper?**
>
> A: We provide statistics on reasoning trace length in Figure 10 (Appendix B.3: Episode Details). For the MATH dataset, the expert traces have an average of 8.42 steps, with a minimum of 1 and a maximum of 228, and a standard deviation of 11.46. For the NuminaMath-CoT dataset, the average is 7.08 steps, with a minimum of 1, maximum of 97, and a standard deviation of 5.40. While reasoning trace length varies widely, BREAD is designed to be robust to this variability. BREAD dynamically creates a learning curriculum by adaptively inserting short expert prefixes/hints.
>
> [1] Saunshi, Nikunj, et al. "Reasoning with latent thoughts: On the power of looped transformers." arXiv preprint arXiv:2502.17416 (2025).
>
> [2] Chu, Tianzhe, et al. "Sft memorizes, rl generalizes: A comparative study of foundation model post-training." arXiv preprint arXiv:2501.17161 (2025).
>
> [3] Li, Yuetai, et al. "Small models struggle to learn from strong reasoners." arXiv preprint arXiv:2502.12143 (2025).
>
> [4] Yu, Qiying, et al. "Dapo: An open-source llm reinforcement learning system at scale." arXiv preprint arXiv:2503.14476 (2025).
>
> [5] Qu, Yuxiao, et al. "Optimizing test-time compute via meta reinforcement fine-tuning." arXiv preprint arXiv:2503.07572 (2025).
>
> [6] Cetin, Edoardo, Tianyu Zhao, and Yujin Tang. "Reinforcement Learning Teachers of Test Time Scaling." arXiv preprint arXiv:2506.08388 (2025).
>
> [7] Dong, Qingxiu, et al. "Reinforcement Pre-Training." arXiv preprint arXiv:2506.08007 (2025).
>
> [8] Muennighoff, Niklas, et al. "s1: Simple test-time scaling." arXiv preprint arXiv:2501.19393 (2025).
>
> [9] Aggarwal, Pranjal, and Sean Welleck. "L1: Controlling how long a reasoning model thinks with reinforcement learning." arXiv preprint arXiv:2503.04697 (2025).

---

> > ### Comment · Reviewer_iFM6 · 2025-08-06
> >
> > Overall, I find the response satisfactory and believe it strengthens the original submission.

---

### Official Review · Reviewer_mfDd · 2025-07-02

**Clarity:** 4
**Significance:** 3
**Originality:** 1
**Rating:** 4
**Confidence:** 5

**Summary:**

This paper proposes a reinforcement learning (RL)-based approach to address the challenges of insufficient exploration by small models on difficult tasks and distributional shift caused by instruction tuning. The authors leverage partial expert demonstrations as starting points to guide model continuation and optimize exploration. Experimental results indicate that the proposed method improves the performance of reasoning tasks on the Qwen series models. However, the paper exhibits significant shortcomings in terms of methodological novelty, experimental design, and generalizability.

**Questions:**

Please address all the issues mentioned in the Weaknesses section as thoroughly as possible.

**Ethical Concerns:**

["NO or VERY MINOR ethics concerns only"]

**Final Justification:**

Some of my concerns have been addressed—for example, the authors have added comparisons with other similar methods and included results based on the Llama series.

However, some of my questions remain unresolved. The authors also acknowledge that the two core findings of the paper are not unique to small models. While the proposed method is novel when applied to small models, there are still similar approaches in the broader landscape of larger models, which affects the originality of the work. I hope that, when resources permit in the future, the authors can further apply their proposed method to larger models to explore its advantages compared to existing methods in those settings.

Additionally, I do not agree with the authors' justification: “It is worth mentioning that the Qwen model family serves as a well-established testbed for mathematical reasoning tasks and methods, as evidenced by its exclusive use in multiple notable papers [6-10].” My reasons are as follows:

DeepSeek-r1 has also released distilled versions based on other models, and did not solely rely on Qwen for experiments.

Regarding the papers [7-10] cited by the authors, upon my thorough review, I found that none of them have been published in any conferences or journals (only one is reported on a Workshop); they are only available on Arxiv. This means that these papers have not undergone rigorous peer review, so the fact that they exclusively used the Qwen model family for experiments does not justify that it is reasonable to do so.

The existence of the Spurious Reward issue in the Qwen model family raises concerns about works that only use Qwen models for experiments, and this should be taken seriously.

The additional experimental results are promising. I will increase my score. However, I believe it is necessary for the authors to include these experiments in the revised version and to provide a more thorough justification for the differences between the proposed method and existing work. Otherwise, readers may be confused about the novelty of this paper.

**Limitations:**

Some of the limitations faced by this paper have been discussed in the Weaknesses section.

**Quality:**

2

**Strengths And Weaknesses:**

### Strengths
1. The problem addressed in this paper is important. On challenging tasks, models often fail to explore correct solution paths, making training difficult.
2. The paper is clearly written and easy to understand.

### Weaknesses

1. Lack of originality in the core idea: The novelty of the proposed method is questionable. The two challenges identified by the authors—difficulty in exploring correct paths on hard problems and distributional shift caused by SFT—are not unique to small models; these issues also exist for larger models.
Moreover, the idea of using partial demonstrations as starting points to facilitate exploration has already been proposed in recent works, such as R^3 [1], which use expert demonstrations to construct curricula for better exploration.
Earlier work can be traced back to OpenAI’s RL research [3]. R^3 also utilizes demonstration-based continuation and validates its effectiveness on mathematical tasks. The main differences between this paper and existing work appear to be the choice of RL algorithm (GRPO vs. PPO) and the focus on small models in this paper. Additionally, relevant prior work is not adequately cited. The authors should clarify the novelty of their proposed approach.

2. Experiments are conducted only on the Qwen series models, which have been shown to exhibit the “performance improvement under random rewards” phenomenon. For example, Shao et al. [4] demonstrated that Qwen models show significantly greater gains under spurious rewards compared to other models, raising concerns about the generalizability of the results. The authors should include evaluations on additional small models to validate the universality of their method.

3. The two core challenges identified by the authors do not appear to be specific to small models; larger models are also affected. I am curious about the effectiveness of the proposed approach on larger models.

4. Lack of comparison with other methods: The experimental section does not include comparisons with other approaches aimed at improving SLM reasoning capabilities, such as R^3. The authors should compare their method against these baselines to better highlight its advantages.

References:

[1] Xi, Z., Chen, W., Hong, B., Jin, S., Zheng, R., He, W., Ding, Y., Liu, S., Guo, X., Wang, J., & Guo, H. (2024, July). Training Large Language Models for Reasoning through Reverse Curriculum Reinforcement Learning. In International Conference on Machine Learning (pp. 54030-54048). PMLR.

[3] Florensa, Carlos, David Held, Markus Wulfmeier, Michael Zhang, and Pieter Abbeel. "Reverse curriculum generation for reinforcement learning." In Conference on robot learning, pp. 482-495. PMLR, 2017.

[4] Shao, Rulin, Shuyue Stella Li, Rui Xin, Scott Geng, Yiping Wang, Sewoong Oh, Simon Shaolei Du et al. "Spurious Rewards: Rethinking Training Signals in RLVR." arXiv preprint arXiv:2506.10947 (2025).

---

> ### Author Rebuttal · Authors · 2025-07-31
>
> Thank you for your detailed and constructive feedback. Below, we address the comments point by point. We would also be grateful to hear any additional feedback or questions.
>
> **W1.1 & W3: The issues BREAD target also exist for larger models.**
>
> A: Thank you for pointing this out. We agree that the two core issues identified in our paper are not exclusive to small models. However, this is not a weakness of the paper, instead, it highlights the method’s potential applicability across a wide range of settings, making it more broadly useful. Due to limited computational resources, we were not able to run experiments on large models. However, we are open to demonstrate BREAD’s scalability to larger models as a promising future direction.
>
> Meanwhile, we remark that these issues are more acute in small models. Also, BREAD benefits from a strong expert/teacher model to provide high-quality traces (human experts may not readily be available). For this reason, our experiments prioritize SLMs, where the impact of BREAD is most clearly observable. Overall, our focus on SLMs is not a limitation, but a significant contribution in itself for several reasons. SLMs are increasingly important for cost-sensitive deployments, such as on-device reasoning and agentic AI [5]. Most reasoning benchmarks show a large performance gap between SLMs and LLMs. Many pipelines that work well for large models fail to scale down to smaller ones. So, improving reasoning in SLMs remains a central open problem.
>
> **W1.2: Novelty concerns and comparison with suggested papers.**
>
> A: Thank you for pointing out connections to previous papers. We will cite the papers and include a discussion in our revised version. We will also add R^3 as a baseline. The results and the setting are provided in the response to W4.
>
> We agree that BREAD shares conceptual similarities with [1,3]. All methods aim to mitigate the challenge of sparse rewards by starting training closer to successful endpoints.
>
> But there are several key differences between BREAD and previous papers:
>
> (1) Dynamic vs. fixed hint: Previous works use a manually defined schedule to adjust the prefix (hint) ratio during training, which requires careful tuning. In contrast, BREAD dynamically selects the prefix length for each question at each iteration, using a binary search to automatically adapt the difficulty.
>
> (2) Use expert traces only when needed: Previous works require full expert traces for all training examples, whereas BREAD adaptively applies hints only when needed, significantly reducing the cost of expert trace generation by human experts or large models.
>
> (3) Cold-start capability: R^3 relies on an initial SFT warm-up phase before applying reverse RL, whereas BREAD can improve performance without requiring any prior SFT initialization. We also provide both empirical and theoretical results showing that SFT initialization can fail, and this phase can be expensive.
>
> (4) Theoretical grounding: Complementing our empirical results and previous works, we provide a Markov chain-based theoretical model to explain why small models could struggle with long reasoning traces, why GRPO without hints can perform poorly, and how BREAD mitigates the sparse reward issues.
>
> These differences show the novelty and broader applicability of BREAD, especially in settings where expert traces are costly and model capacity is limited.
>
> **W2: Shao et al. raise concerns about the generalizability of the results evaluated on Qwen family models.**
>
> A: Thank you for the question. The main message in Shao et al. [4] is not contradictory with ours: They argue that pure RLVR cannot improve the models’ ability, but makes the latent ability in the base models show up. That’s exactly why we should utilize expert traces in BREAD so that new knowledge can be infused during RL training to truly improve the models’ reasoning ability. Notably, [4] states a maximum accuracy of 78.5% on MATH using Qwen2.5-7B, whereas our paper achieves 84.3% with BREAD using the smaller Qwen2.5-3B under the same number of iterations. So it also doesn’t contradict BREAD’s performance benefits, as BREAD goes strictly beyond RLVR with both spurious and true labels.
>
> Additionally, we evaluate BREAD by training LLama3-1B on GSM8k dataset. BREAD achieves meaningful improvements over vanilla GRPO and noticeably outperforms SFT+GRPO, while also reducing the cost of expert trace generation by selectively using hints rather than full traces for every example.
>
> |GRPO|SFT+GRPO|BREAD|
> |-|-|-|
> |9.0%|36.6%|39.7%|
>
> It is worth mentioning that the Qwen model family serves as a well-established testbed for mathematical reasoning tasks and methods, as evidenced by its exclusive use in multiple notable papers [6-10]. However, we are grateful for the reviewer's recommendation to use additional models and will provide additional evaluations in the final version.
>
> **W4: Lack of comparison with other methods.**
>
> A: Thank you for the suggestion. We ran additional experiments for R^3 as the baseline on MATH. For fair comparison, we deploy the same R^3 on GRPO and train for 300 steps in total for both methods.
>
> BREAD demonstrates clear improvements in both accuracy and sample efficiency. Unlike R^3, which requires full expert traces for every example, BREAD learns effectively using only a small fraction of expert traces. This is important because collecting full reasoning traces is prohibitively expensive. It requires either human labeling or generation by large models such as DeepSeek-R1, whose single forward pass already costs more FLOPs than 25 RL training steps with 8 rollouts. BREAD’s ability to reduce this cost while improving performance highlights its practical value.
>
>
> |Model|Method|Acc|Expert Trace Usage|
> |-|-|-|-|
> Qwen-1.5B-Instruct|GRPO|59.0%|/|
> Qwen-1.5B-Instruct|BREAD|78.8%|27.8%|
> Qwen-1.5B-Instruct|R^3| 70.2%|100%|
> Qwen-3B-Instruct|GRPO|68.1%|/|
> Qwen-3B-Instruct|BREAD|84.3%|19.1%|
> Qwen-3B-Instruct|R^3|74.6%|100%|
>
>
> [1] Xi, Zhiheng, et al. "Training large language models for reasoning through reverse curriculum reinforcement learning." arXiv preprint arXiv:2402.05808 (2024).
>
> [3] Florensa, Carlos, et al. "Reverse curriculum generation for reinforcement learning." Conference on robot learning. PMLR, 2017.
>
> [4] Shao, Rulin, et al. "Spurious rewards: Rethinking training signals in rlvr." arXiv preprint arXiv:2506.10947 (2025).
>
> [5] Belcak, Peter, et al. "Small Language Models are the Future of Agentic AI." arXiv preprint arXiv:2506.02153 (2025).
>
> [6] Guo, Daya, et al. "Deepseek-r1: Incentivizing reasoning capability in llms via reinforcement learning." arXiv preprint arXiv:2501.12948 (2025).
>
> [7] Cetin, Edoardo, Tianyu Zhao, and Yujin Tang. "Reinforcement Learning Teachers of Test Time Scaling." arXiv preprint arXiv:2506.08388 (2025).
>
> [8] Dong, Qingxiu, et al. "Reinforcement Pre-Training." arXiv preprint arXiv:2506.08007 (2025).
>
> [9] Muennighoff, Niklas, et al. "s1: Simple test-time scaling." arXiv preprint arXiv:2501.19393 (2025).
>
> [10] Aggarwal, Pranjal, and Sean Welleck. "L1: Controlling how long a reasoning model thinks with reinforcement learning." arXiv preprint arXiv:2503.04697 (2025).

---

> > ### Comment · Reviewer_mfDd · 2025-08-04
> > **Response to authors**
> >
> > Thank you for your response and the additional experiments. Some of my concerns have been addressed—for example, the authors have added comparisons with other similar methods and included results based on the Llama series.
> >
> > However, some of my questions remain unresolved. The authors also acknowledge that the two core findings of the paper are not unique to small models. While the proposed method is novel when applied to small models, there are still similar approaches in the broader landscape of larger models, which affects the originality of the work. I hope that, when resources permit in the future, the authors can further apply their proposed method to larger models to explore its advantages compared to existing methods in those settings.
> >
> > Additionally, I do not agree with the authors' justification: “It is worth mentioning that the Qwen model family serves as a well-established testbed for mathematical reasoning tasks and methods, **as evidenced by** its **exclusive use** in multiple notable papers [6-10].” My reasons are as follows:
> > 1. DeepSeek-r1 has also released distilled versions based on other models, and did not solely rely on Qwen for experiments.
> >
> > 2. Regarding the papers [7-10] cited by the authors, upon my thorough review, I found that none of them have been published in any conferences or journals (only one is reported on a Workshop); they are only available on Arxiv. This means that these papers have not undergone rigorous peer review, so the fact that they exclusively used the Qwen model family for experiments does not justify that it is reasonable to do so.
> >
> > 3. The existence of the Spurious Reward issue in the Qwen model family raises concerns about works that only use Qwen models for experiments, and this should be taken seriously.

---

> ### Comment · Reviewer_mfDd · 2025-08-04
> **Response to authors (2/2)**
>
> Based on the authors' response, I will raise my score. However, I still recommend that the authors carefully consider these comments.

---

> ### Author Response · Authors · 2025-08-05
>
> Thank you for engaging with and acknowledging our response and the additional experiments we have provided.
>
> We fully agree that broader validation is important and are pleased to report that our method remains effective when applied to LLaMA models. In particular, our experiments in previous response with the LLaMA3-1b model show notable gains over baselines. We also conducted additional experiments on the LLaMA3.2-3b-Instruct model. The results are consistent with Qwen experiments. For the final manuscript, we will incorporate at least one more model family, such as Phi3 or Phi4 models from Microsoft. To make sure our comparison is fair, we follow the LLaMA3 technical report, which uses a 0-shot CoT prompt. Specifically, because the technical report does not provide a detailed prompt, we use a general and intuitive 0-shot CoT prompt and provide it here to make sure our result is reproducible:
> ```
> messages = [
>   {“role”: “system”, “content”: “You are a helpful assistant that reasons step by step to solve complex problems.“},
>   {“role”: “user”, “content”: QUERY + “Let’s think step by step.“}
> ]
> ```
> The result are for the LLaMA3.2-3b-Instruct model and are as follows:
>
> (1) On the full MATH dataset:
>
> |GRPO|SFT+GRPO|BREAD|
> |-|-|-|
> |43.6%|45.9%|48.2%|
>
> (2) We also identify a hard subset of MATH dataset using similar methodology as described in the paper (Figure 7). Training over this dataset yields:
>
> |GRPO|SFT+GRPO|BREAD|
> |-|-|-|
> |0.0%|3.0%|6.0%|
>
> These experiments support that the method generalizes beyond the Qwen family and is not model-specific, and the effectiveness is more significant when the task is harder.
>
> While our focus is on small models due to compute constraints, we are committed to scaling our experiments to larger models and other model families in future work to further validate the general efficacy of our approach.
>
> Thank you again for your constructive feedback.

---

> > ### Comment · Reviewer_mfDd · 2025-08-07
> > **Response to authors**
> >
> > Thank you for conducting the additional experiments. The new experimental results are promising. I will increase my score. However, I believe it is necessary for the authors to include these experiments in the revised version and to provide a more thorough justification for the differences between the proposed method and existing work. Otherwise, readers may be confused about the novelty of this paper.

---

> > > ### Author Response · Authors · 2025-08-07
> > >
> > > We sincerely thank you for your constructive feedback and for recognizing the value of the additional experiments.
> > >
> > > We agree that including these results in the revised version is important, and we will ensure they are integrated along with a more comprehensive discussion of related work. In particular, we will clarify the differences between our method and prior approaches to better highlight the novelty and contributions of our work.
> > >
> > > Thank you again for your thoughtful suggestions.

---

### Official Review · Reviewer_vjLr · 2025-07-02

**Clarity:** 3
**Significance:** 2
**Originality:** 2
**Rating:** 4
**Confidence:** 4

**Summary:**

This paper proposes BREAD: a method for improving GRPO training of small language models on hard reasoning problems using backplay.

**Questions:**

- Is there a correlation between a model's averaged pass@1 accuracy on a question and the amount of hint that needs to be shown before reaching the correct answer? Could this be leveraged to warm-start the binary search?

**Ethical Concerns:**

["NO or VERY MINOR ethics concerns only"]

**Final Justification:**

While BREAD demonstrates some improvement in the reasoning abilities of small models, the improvement is not significant (>5%) and has limited novelty (drawing heavily from prior ideas in backplay).

**Limitations:**

Yes

**Quality:**

2

**Strengths And Weaknesses:**

**Strengths:**

- The paper is clear and easy to follow
- The method demonstrates strong results for SLMs on hard problem sets

**Weaknesses**:

- BREAD is essentially an application of backplay [1,2] to reasoning problems. Despite this, no references on backplay are cited or discussed.
- Evaluation is done only on the Qwen2.5 model series. A more thorough evaluation investigating what characteristics of a model necessitate backplay supervision would be interesting.

Missing references:
- [1] Salimans, Tim and Richard J. Chen. “Learning Montezuma's Revenge from a Single Demonstration.” ArXiv abs/1812.03381 (2018): n. pag.
- [2] Resnick, Cinjon et al. “Backplay: "Man muss immer umkehren".” ArXiv abs/1807.06919 (2018): n. pag.

---

> ### Author Rebuttal · Authors · 2025-07-31
>
> Thank you for the detailed and constructive feedback. Here, we answer the questions and address the concerns. We would be grateful to address further questions during the discussion period.
>
> **W1: lack of references of backplay.**
>
> A: Thank you for pointing this out. We agree that BREAD shares conceptual similarities with Backplay [1,2]. Both methods aim to mitigate the challenge of sparse rewards by starting training closer to successful endpoints and gradually increasing difficulty. We will cite these works and include a discussion in the Related Work section.
>
> Also, we would like to emphasize several key differences between BREAD and classical Backplay. While Backplay typically replays from fixed intermediate states and often requires delicate tuning, BREAD dynamically selects the hint ratio for each question during training, enabling adaptive difficulty adjustment without additional tuning overhead. This forms a curriculum tailored to each query, and offers significant computational efficiency.
>
> Moreover, unlike Backplay, BREAD does not require expert traces for all training samples. This is important as generating expert traces is often prohibitively expensive for reasoning tasks(e.g., requiring human annotation or large model inference).
>
> **W2: What characteristics of a model make BREAD necessary or particularly beneficial?**
>
> A: Thank you for this question. We provide a general discussion of this topic in Section 5 (Discussion and Limitations), and we will expand on it further in the revised version.
>
> BREAD is motivated by the observation that certain model characteristics make it challenging to learn complex reasoning behaviors through standard SFT or the model’s initialization has a very small likelihood of success so RL alone fails. In such cases, curriculum-based or backplay-style learning becomes particularly helpful. In particular, we believe BREAD is most beneficial under the following conditions:
>
> 1. Limited Model Scale: If the base model is strong enough to generate at least one correct rollout per query, BREAD becomes identical to the vanilla GRPO. However, for smaller or weaker models that cannot do so reliably, BREAD offers a strict advantage. So SLMs, that struggle to imitate or generate correct rollouts, are a key beneficiary.
>
> 2. Weak Prior Reasoning Ability: Models that lack training on structured reasoning tasks have difficulty in producing stable and correct reasoning traces during RL exploration. BREAD helps stabilize training by anchoring trajectories to expert traces. For instance, in our experiments, vanilla GRPO based on Qwen2.5-1.5B-Instruct shows limited improvement and is much worse than BREAD (Table 2). Whereas DeepScaleR-1.5B-Preview[5] achieves stronger results using DeepSeek-Distill-1.5b[3], which has the same architecture, but was obtained after careful supervised training. BREAD mitigates the need for this expensive supervised training stage. So BREAD is especially helpful when the base model starts from a weaker reasoning prior.
>
> 3. Hint-Responsiveness: BREAD is most effective when the student model can utilize partial hints to guide its reasoning. However, in some extreme cases, a model may fail to benefit from expert traces even when provided in full, due to its inability to understand and follow complex traces. For instance, Llama‑3.2‑1B‑Instruct[6] achieves 6.8% accuracy on MATH‑500, and this improves marginally to 7.3% with hints. Since the model cannot meaningfully use the hints, BREAD provides limited benefit (achieves accuracy 8.9%) in such scenarios.
>
> **Q1: Can we use the model's average accuracy on a question to warm-start or guide the branching point (e.g., binary search) more efficiently?**
>
> A: Thank you for raising this excellent suggestion. We do expect a correlation: questions with high pass@1 accuracy likely require little or no hinting, while harder questions benefit from longer anchored prefixes (i.e., branching closer to the answer). Our algorithm design reflects this intuition. However, in our current implementation, we prioritize sample efficiency and we only do binary search for queries where the model fails to generate any correct rollout (pass@1 = 0). This helps avoid generating expensive expert traces for easier questions. We discuss how expensive it is to generate the trace in paragraph "BREAD improves sample efficiency during training". But we agree that incorporating accuracy to warm-start the binary search could improve efficiency if we want to binary search on queries with different accuracy, and we see this as a promising direction for future work.
>
> [1] Salimans, Tim, and Richard Chen. "Learning montezuma's revenge from a single demonstration." arXiv preprint arXiv:1812.03381 (2018).
>
> [2] Resnick, Cinjon, et al. "Backplay:" man muss immer umkehren"." arXiv preprint arXiv:1807.06919 (2018).
>
> [3] Guo, Daya, et al. "Deepseek-r1: Incentivizing reasoning capability in llms via reinforcement learning." arXiv preprint arXiv:2501.12948 (2025).
>
> [4] Liu, Aixin, et al. "Deepseek-v3 technical report." arXiv preprint arXiv:2412.19437 (2024).
>
> [5] Luo, Michael, et al. DeepScaleR: Surpassing O1-Preview with a 1.5B Model by Scaling RL. Notion Blog, 2025, https://pretty-radio-b75.notion.site/DeepScaleR-Surpassing-O1-Preview-with-a-1-5B-Model-by-Scaling-RL-19681902c1468005bed8ca303013a4e2.
>
> [6] Meta AI. Llama‑3.2‑1B‑Instruct. Meta Llama 3.2 model card, Hugging Face, 25 Sept. 2024.

---

> > ### Comment · Reviewer_vjLr · 2025-08-05
> >
> > Thank you for the detailed response. I will maintain my positive score.

---

### Official Review · Reviewer_YNri · 2025-07-03

**Clarity:** 3
**Significance:** 3
**Originality:** 3
**Rating:** 5
**Confidence:** 3

**Summary:**

This work introduces new GRPO-like algorithms for training Small Reasoning models. The authors leverage the observation that small models required longer traces to match larger models, hence, SFT might hurt performance. Moreover, because GRPO relies on having a good initialization from SFT, subsequent RL training can more likely fail.  Hence, the authors designed BREAD, an algorithm that leverages the expert demonstrations (reasoning traces) to adaptively choose how much of the expert trace to prepend to achieve a threshold success rate of the sampled group. This allows the model to obtain a kind of curriculum that improves RL training and bypasses the SFT training phase. The empirical evaluation successfully shows that their algorithm improves over all baselines.

**Questions:**

- I appreciate the toy model provided that serves to motivate the algorithm design. I’m mostly interested in understanding how much we can trust it. Where do you get the motivation of modelling the limited capacity of small models by limiting the learnable graphs in the Markov chain?
- Can we apply BREAD for general LLM reasoning or does it only apply to SLMs? Do you expect that without the bottleneck of the model capacity, BREAD could still provide an advantage over GRPO?
- As I understand it, the expert guidance is incorporated by modifying the search process. Effectively changing the policy \pi that is being executed, doesn’t this introduce off-policy issues?

**Ethical Concerns:**

["NO or VERY MINOR ethics concerns only"]

**Final Justification:**

The authors addressed most of my concerns effectively and provided further insight about the method.
Only the off-policy issues of the algorithm remained unresolved but I believe they do not reduce the merit of the current method. Moreover, the authors have provided further baselines and evidence of the empirical performance of their method during the rebuttal period that strengthens their evaluation.

**Limitations:**

yes

**Paper Formatting Concerns:**

No major formatting issues.

**Quality:**

3

**Strengths And Weaknesses:**

Strengths:
- The paper is very well written and provides a clear exposition of the motivation and design of BREAD;
- The authors include a reasonable set of baselines, including naive variations of GRPO, and show that their algorithm improves over all of them.
- The authors provide experiments that allows to further understand the impact of their algorithm on compute, sample efficiency and effect of the hardness of the tasks/questions

Weaknesses:
- The toy model builds a nice intuition for the problem motivation but it may be too limited. Though I understand this may not be the main contribution of this work.
- In terms of the BREAD objective, I believe there might be a problem with off-policy. See questions.

---

> ### Author Rebuttal · Authors · 2025-07-31
>
> Thank you for your encouraging feedback on our manuscript. We’re grateful for your recognition of our approach and your constructive comments. Below, we respond to your questions and concerns. We would be grateful to address further questions during the discussion period.
>
> **Q1: What motivates modeling the limited capacity of small models by limiting the learnable graphs in the Markov chain?**
>
> A: Our motivation comes from three insights: (1) Recent studies[1] show that an L-times deeper LLM can simulate L chain-of-thought steps of a smaller model. This implies that for a student model with limited depth/capacity, a dense reasoning trace from a stronger expert must be broken down into shorter, more manageable segments to enable learning via SFT. (2) Compositional nature of reasoning: Reasoning tasks are inherently compositional and typically require multi-step chain-of-thought processes to arrive at a correct solution [2]. Small models often struggle to learn such long trajectories end-to-end. (3) Markov chains are more amenable to theoretical analysis while being representative of sequential/language data [3-7].
>
> **Q2: Can BREAD generalize beyond SLMs?**
>
> A: Yes, BREAD is a general method that is not limited by model size. We thank the reviewers for raising this point, and we are open to demonstrate BREAD’s scalability to larger models as a future direction. This highlights the method’s potential applicability across a wide range of settings, making it even more broadly useful.
>
> BREAD targets the fundamental limitations of SFT and RL. SFT often struggles to teach hard-to-imitate traces, and standard RL often suffers from sparse rewards. Although these problems are more noticeable in small models, high-capacity models can also struggle with similar difficulties, particularly when dealing with complex reasoning tasks where high-quality traces are hard to generate and the RL reward is sparse due to low probability of success. For example, Qwen2.5-7B-MATH, the accuracy drops from 13.3% to 6.67% on AIME from 15.1% to 5.6% on GPQA after SFT with S1K dataset. Qwen2.5-32B-Instruct achieves only 10% accuracy on AIME, which improves to 37% after fine-tuning on the S1K dataset. However, even at 37% accuracy, reward signals in RL remain sparse(19.5% of samples all 8 rollouts fail). BREAD addresses this by adapting problem difficulty through a curriculum learning that interpolates between expert (offline) traces and self-generated (online) rollouts. We therefore believe BREAD can help in large-model settings, particularly for tasks involving long reasoning chains or limited supervision. But because of the computational source constraint, it’s hard for us to show results on larger models. We will continue this as future work.
>
> **Q3: Does BREAD introduce off-policy learning issues?**
>
> A: The expert traces are used to guide the trajectory generation, not to replace or bypass the model’s policy. In detail, BREAD samples rollouts by interpolating between the expert traces and the self-generated traces to adapt difficulty. However, the policy gradient update is always computed using the log-probabilities of the model’s own trace(online part). This ensures that the learning signals remain on-policy, even though the rollout paths may be partially guided by the expert. Therefore, BREAD benefits from expert traces without breaking the assumptions of policy gradient methods.
>
> [1] Saunshi, Nikunj, et al. "Reasoning with latent thoughts: On the power of looped transformers." arXiv preprint arXiv:2502.17416 (2025).
>
> [2] Wei, Jason, et al. "Chain-of-thought prompting elicits reasoning in large language models." Advances in neural information processing systems 35 (2022): 24824-24837.
>
> [3] Zekri, Oussama, et al. "Large language models as markov chains." arXiv preprint arXiv:2410.02724 (2024).
>
> [4] Makkuva, Ashok Vardhan, et al. "Attention with markov: A curious case of single-layer transformers." The Thirteenth International Conference on Learning Representations. 2025.
>
> [5] Edelman, Ezra, et al. "The evolution of statistical induction heads: In-context learning markov chains." Advances in neural information processing systems 37 (2024): 64273-64311.
>
> [6] ldiz, M. Emrullah, et al. "From self-attention to markov models: Unveiling the dynamics of generative transformers." arXiv preprint arXiv:2402.13512 (2024).
>
> [7] Kim, Hyunsu, et al. "Parameter Expanded Stochastic Gradient Markov Chain Monte Carlo." arXiv preprint arXiv:2503.00699 (2025).

---

> > ### Comment · Reviewer_YNri · 2025-08-06
> >
> > Thank you for the responses to my questions! I appreciate it.
> > I still believe that the guidance process introduces off-policy issues that might be implicitly handled by the clipping objective of GRPO/PPO. BREAD modifies the trajectories sampled through the search process, effectively modifying policy $\pi$ to a new policy $\pi_b$ that has a higher success rate.
> > However, I can see that empirically this is not affecting the performance. I wonder if taking into account the off-policy issues would further improve performance or the clipping objective is enough. I'd encourage authors to look into this more.

---

> ### Author Response · Authors · 2025-08-06
>
> Thank you for the helpful suggestion. We plan to include additional ablation studies in the next version to address this concern.
>
> (1) We will evaluate and compare unclipped BREAD and SFT + unclipped GRPO, which will help determine whether clipping has a more significant effect on BREAD or not.
>
> (2) Regarding the off-policy issue, we agree with the reviewer that a "more online" version is unlikely to significantly improve over our current algorithm. We will include a more detailed discussion on this point and provide empirical results for a "more online" variant to further support this claim.
>
> Preliminary experiments confirm this expectation. We tested a more online variant on the difficult subset (same setting as Figure 7) and observed only a marginal difference, just a one-question gap, which may be due to random variation.
> In this "more online" variant, rather than relying entirely on fixed expert traces, we have declared any successful trace generated by BREAD as the new expert trace. This dynamic update mechanism aims to keep the training more aligned with the current policy, as long as the model continues to make progress by producing successful solutions.

---

### Note · Authors · 2025-08-13

We sincerely thank the reviewers for their time and thoughtful engagement during the discussion period. The exchange has been highly constructive, allowing us to address all concerns in detail. We are confident that the feedback received will strengthen the final manuscript, and we truly appreciate the reviewers’ time and effort to improving this work.

---

### Decision · Program_Chairs · 2025-09-17

**Decision:**

Accept (poster)

**Comment:**

The authors propose a method named BREAD to densify the reward signal when learning to improve SLM's reasoning with the help of a larger expert model. The method is well motivated. When self-generated traces of SLM fail, BREAD adaptively inserts short expert prefixes/hints and hence densify the reward signal. A toy example illustrates and justifies the idea well, and the experimental results demonstrate the effectiveness of BREAD. Some issues (e.g. experiments on models other than Qwen; off-policy vs on-policy) were raised by the reviewers, and most of them were well addressed during the rebuttal. The reviewers are all positive on the contribution of the paper. Overall, it is a well-motivated, simple&effective, well-justified solution to the investigated problem. I hence recommend acceptance. In the meantime, I would suggest the authors to further strength the paper based on the rebuttal.